# Evolutionary design of explainable algorithms for biomedical image segmentation

Kévin Cortacero [1,2,3], Brienne McKenzie[1,2,3], Sabina Müller[1,2,3], Roxana Khazen[1,2,3], Fanny Lafouresse [1,2,3], Gaëlle Corsaut[1,2,3], Nathalie Van Acker[4], François-Xavier Frenois[4], Laurence Lamant[4], Nicolas Meyer[5], Béatrice Vergier[6,7], Dennis G. Wilson[8], Hervé Luga[8], Oskar Staufer[9,10], Michael L. Dustin [9], Salvatore Valitutti [1,2,3,4,11] ✉ & Sylvain Cussat-Blanc [8,11] ✉

An unresolved issue in contemporary biomedicine is the overwhelming number and diversity of complex images that require annotation, analysis and interpretation. Recent advances in Deep Learning have revolutionized the field of computer vision, creating algorithms that compete with human experts in image segmentation tasks. However, these frameworks require large human-annotated datasets for training and the resulting "black box" models are difficult to interpret. In this study, we introduce *Kartezio*, a modular Cartesian Genetic Programming-based computational strategy that generates fully transparent and easily interpretable image processing pipelines by iteratively assembling and parameterizing computer vision functions. The pipelines thus generated exhibit comparable precision to state-of-the-art Deep Learning approaches on instance segmentation tasks, while requiring drastically smaller training datasets. This Few-Shot Learning method confers tremendous flexibility, speed, and functionality to this approach. We then deploy Kartezio to solve a series of semantic and instance segmentation problems, and demonstrate its utility across diverse images ranging from multiplexed tissue histopathology images to high resolution microscopy images. While the flexibility, robustness and practical utility of Kartezio make this fully explicable evolutionary designer a potential game-changer in the field of biomedical image processing, Kartezio remains complementary and potentially auxiliary to mainstream Deep Learning approaches.

Contemporary imaging techniques in biology and medicine produce a tremendous quantity of images, which must be analyzed to provide meaningful interpretations of biological and clinical processes. The scale, context, and resolution of such images is highly variable, ranging from super-resolution and electron microscopy in biology to whole-tissue imaging in clinical pathology. Generating "actionable intelligence" from such imaging data, using either human or automated analysis of images, requires quantitative knowledge extraction[1]. The bottleneck of the knowledge extraction process has historically been the human power required for image evaluation, annotation, and interpretation, all of which can be time-consuming, prone to inter-human variability, and subject to error. Moreover, human nature

imposes a subjective framework of reasoning that can affect the overall objectivity of the knowledge extraction process and the subsequent validation or invalidation of a given hypothesis. The development of sophisticated automated image analysis pipelines that preserve the nuances of human insight, while circumventing the abovementioned limitations, is a formidable challenge in contemporary biomedicine.

To keep pace with the development of increasingly sophisticated imaging techniques, engineers and researchers have designed a variety of image processing filters and pipelines that assist experimentalists and clinicians in extracting quantitative information from their digital images. Nevertheless, filters and pipelines need a certain level of expertise to be used optimally. Artificial Intelligence (AI) and more specifically Deep Learning (DL) approaches have since provided a great step forward in image processing and revolutionized the field of Computer Vision (CV). This is particularly evident in tasks such as Semantic Segmentation (SS), which clusters together parts of an image belonging to the same object class, and Instance Segmentation (IS), a complex task that additionally involves the demarcation of overlapping or interacting objects even if they belong to the same class.

AI was proven to be competitive with human expertise in medical image analysis within the fields of cardiology, dermatology, ophthalmology, radiology and pathology among others[2,3]. To date, these approaches have primarily been based on artificial Deep Neural Networks (DNNs), which are conceived to work at the pixel level to generate filters and pipelines from scratch without building on previous human knowledge and experience. This requires huge computational efforts to provide satisfactory solutions. DNNs are composed of thousands of simple computational units, the neurons, which are connected in a complex network of weighted interactions[4]. Designing the architecture of such a network is a key step before training. Because the subsequent training of an DNN to solve a given task requires the optimization of millions of parameters, DNNs require huge computational efforts to provide satisfactory solutions. Despite important advances in eXplainable AI (XAI)[5,6], DNNs have generally been difficult for humans to analyze and interpret, leading to the definition of these models as "black-boxes"[5]. As such, there is substantial interest in developing more explainable "white-box" methods to complement existing DNN approaches in medical and biological image analysis.

Herein, we propose an innovative approach to image analysis that tackles the important challenges described above while remaining fully explainable and interpretable to humans. Our approach, named "Kartezio", is based on the evolutionary design of pipelines assembled from pre-existing human-engineered filters. This is accomplished by using a specific type of Genetic Programming (GP)[7] algorithm, known as Cartesian Genetic Programming (CGP)[8–10], to generate image processing pipelines, known as CGP for Image Processing (CGP-IP)[11–13]. As a case study for this powerful approach, we deployed Kartezio to solve different IS and SS challenges, thematically centered around the field of tumor immunotherapy.

Proposed by Miller in 1999[10], CGP has been successfully used in multiple variants (Mixed Types, Recurrent, Self-Modifying, etc. as reviewed in[14]) for logical circuit generation[9], symbolic regression[15], agent control[16], neural architecture search[17,18] and simple image processing tasks such as noise reduction[19] and SS[11,12] but to date the application of this powerful approach to IS in biomedicine has been limited. By merging CGP-based IS with CV measures and classical unsupervised machine learning (ML), we designed a highly effective AI solution to extract quantitative information from immunofluorescent (IF) and immunohistochemical (IHC) images. Our CGP-based strategy is: (i) trainable on small datasets, (ii) transparent and interpretable by humans, and (iii) capable of integrating human knowledge. The strategy presented in this manuscript (summarized in Supplementary Fig. 1) represents a substantially new conceptual framework for the design of image processing pipelines, and offers a novel approach to

tackle the challenge of image analysis in contemporary medical and biological sciences.

## Results

### Evolving programs for instance segmentation with CGP

GP is based on the artificial evolution of a population of syntactic graphs composed of mathematical functions that, when executed, process given inputs to produce an expected output[7]. While GP directly works on the optimization of syntactic graphs, CGP considers a specific encoding of this graph within an integer-based genotype, also called genome (Fig. 1a, see Material and Methods). In this paper, we developed a modular framework called *Kartezio* to expand CGP into IS tasks and to confer enhanced versatility to its application, thus revitalizing the CGP approach to iteratively evolve image segmentation pipelines, as illustrated in Supplementary Movie 1 (images reproduced with permission). To this end, we first constituted a library of 42 functions specifically designed for image processing (full library is provided in Supplementary Table 1). In addition, we introduced the notion of non-evolvable nodes which are functions not subjected to optimization of the syntactic graph (Fig. 1). The objective of these two initial steps was to introduce human knowledge about the input images, expected outputs and, if requested, additional known operators into CGP-generated pipelines. With these adaptations, we reinvigorated the CGP paradigm to make it more versatile and introduced (i) non-evolvable preprocessing nodes to transform input images (e.g. transmission light images, fluorescent images, IHC tissue images, etc.) into the most appropriate format for the expected outcome, and (ii) appropriate endpoints (such as Watershed Transform[20,21] or Circle Hough Transform[22]) which can strongly reduce the search space to perform IS, similarly to what is done in many current DNN-based methods[23–27]. Together the decoded genotype and the non-evolvable nodes will produce executable image processing pipelines (Fig. 1b).

In the next section, we compare our CGP approach to state-of-the-art DL techniques applied to cell IS within microscopy and histochemical images.

### Evaluation of Kartezio performance versus state-of-the art Deep Learning models

To evaluate the performance of this approach against state-of-the-art DL models, we compared Kartezio to Cellpose[23], a recently released DL approach for IS in biology with both specialist and generalist models, as well as to two previously released approaches, Mask R-CNN[25] and StarDist[24]. As described below, we first evaluated Kartezio on the Cellpose dataset (originally derived from Cell Image Library)[28] and compared it to the reported performance of the Cellpose specialist model (Fig. 2a, b)[23]; we then assessed both Kartezio and the Cellpose generalist model side-by-side on our own recently published dataset (Fig. 2c, d)[29].

In their initial study, Stringer et al. introduced the Cellpose model to perform IS on diverse types of cells or nuclei; this generalist model was trained on over 70,000 segmented objects[23]. The authors also proposed a distinct specialist model, trained on in vitro cultured neurons (individual channels from a representative image are displayed in Supplementary Fig. 2a; reproduced with permission); this dataset was composed of 89 training images and 11 test images[23,28]. Since Kartezio is a modular framework for generating specialist models, we chose to initially compare our approach to the Cellpose specialist model, and accordingly utilized the 89-image training dataset and 11-image test dataset previously used in the generation of the Cellpose specialist model[23,28].

We utilized randomly selected subsets of these 89 images (ranging from 1 to 89 images) to train Kartezio from scratch and subsequently evaluated its performance on the previously unseen 11-image Cellpose test dataset. Figure 2a and Supplementary Table 2 show the

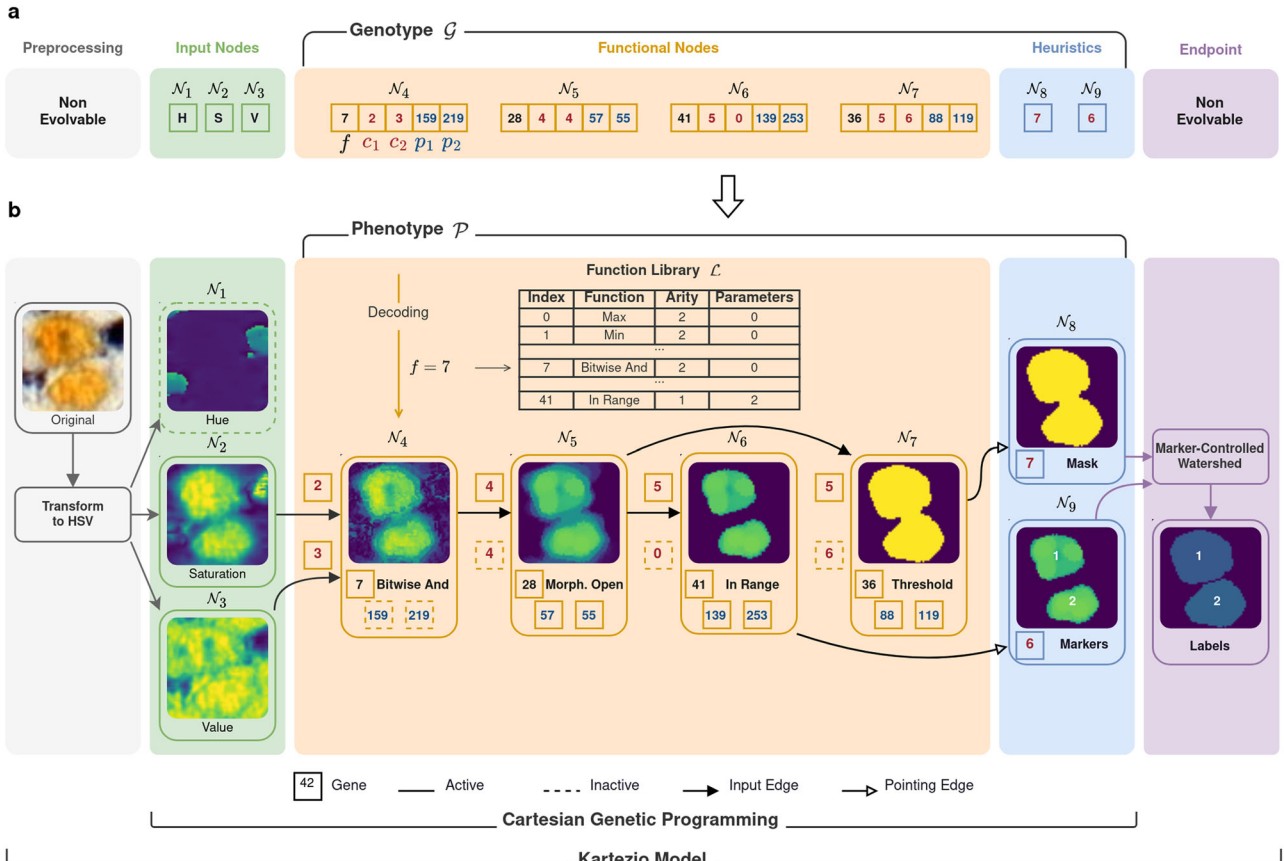

**Fig. 1 | Architecture of a Kartezio-generated model: a fully explainable and transparent algorithm for Instance Segmentation. a**, **b** An illustrative example of a pruned model trained using Kartezio on a melanoma tissue nodule dataset derived from a previously published melanoma cohort (see Supplementary Fig. 2b)[29]. From left to right, preprocessing is represented in *gray*; input nodes are shown in *green*; functional nodes are shown in *orange*; heuristics are shown in *blue*; and endpoint is shown in *purple*. **a** The genotype $\mathcal{G}$ is a sequence of integers known as genes (each one represented as a box containing a single integer) that are organized into nodes, $\mathcal{N}_i$. A node is composed of a single function $f$ (values shown in black) drawn from the function library $\mathcal{L}$, at least one connection (values shown in red, in this example, $c_1$ and $c_2$), and optional parameters (values shown in blue, in

this example, $p_1$ and $p_2$). Nodes are categorized as either functional nodes (in this example, $\mathcal{N}_{4\rightarrow7}$) or outputs (in this example, $\mathcal{N}_{8\rightarrow9}$). In this example, three entry channels ($\mathcal{N}_{1\rightarrow3}$), four functional nodes ($\mathcal{N}_{4\rightarrow7}$) and two outputs ($\mathcal{N}_{8\rightarrow9}$) are shown. **b** The phenotype $\mathcal{P}$ decoded from the genotype described in a) takes the form of a directed acyclic graph. To obtain the active graph (*solid lines*), only the genes that are directly or indirectly connected to outputs are considered. The last operation of the pipeline, called the Endpoint, adds a non-evolvable user-chosen function (in this case, the Marker-Controlled Watershed for Instance Segmentation) that creates an intentional bottleneck to drive the heuristics optimization. While the Endpoint is part of the evaluation of the genotype, it is not subjected to evolution and is chosen by the user depending on the task to solve.

scores obtained on the 11-image test dataset after first training Kartezio on datasets of different sizes (n = 35 experiments per dataset size). Dotted lines indicate the results obtained by the other three methods as reported in Stringer et al.[23], when trained on all 89 images. It is interesting to note that Kartezio frequently outperformed Mask R-CNN[25] and StarDist[24] even when trained on much smaller datasets (e.g. on average, Kartezio matched Mask R-CNN when trained on as few as 6 images, and matched StarDist when trained on only 3 images). This illustrates the exceptional capacity of Kartezio to extrapolate from a very small set of training images (a property known as "few-shot learning") and yet achieve comparable or superior accuracy compared to state-of-the-art DL approaches. Our results also show that the best score obtained by Kartezio was 0.89, a value comparable to the reported best performance of Cellpose (0.91). Of note, increasing the size of the training set in 10-image increments from 30 to 80 images did not perceptibly change Kartezio's performance.

The capacity of Kartezio to extrapolate from only a few training events was also evident when we compared the scores obtained on training subsets with those obtained on the 11-image test dataset. In this analysis, a score of 0 is optimal as it shows no decrease in accuracy when transitioning from the training to the test set. As shown in Fig. 2b, the difference of mean average precision (AP) between the training

and test sets was close to 0 when only 9-10 images were used for training. These data indicate that Kartezio generates pipelines of comparable precision to state-of-the-art DL approaches but with a drastic reduction in the size of the training dataset. Interestingly, Kartezio generalized well on this dataset as the training versus testing scores were comparable (Supplementary Table 2).

In a second exercise, we compared the performance of Kartezio to the Cellpose generalist model (CPx) in the context of identifying individual melanoma nuclei within complex real world histopathology images (Fig. 2c, d). The original dataset, consisting of tissue samples from melanoma patients and the associated clinical data, has been previously described[29]. In the present study, we employed 12 histopathology images for Kartezio training, and a further 10 images for Kartezio and Cellpose side-by-side comparison (cropped as shown in Fig. 2d). As depicted in Fig. 2c, we also used this opportunity to evaluate different non-evolvable nodes in the Kartezio pipeline: RGB versus HSV versus HED in the pre-processing node, and Marker Controlled Watershed (MCW) versus No Endpoint in the endpoint node (n = 10 experiments per condition). Individual RGB, HSV, and HED channels are shown in Supplementary Fig. 2b. All six CGP-based approaches significantly outperformed CPx (AP50 = 0.605) on this dataset (RGB$_{MCW}$ $p$ = 0.000030, $t$ = 7.697; RGB$_{No\ Endpoint}$ $p$ = 0.000049,

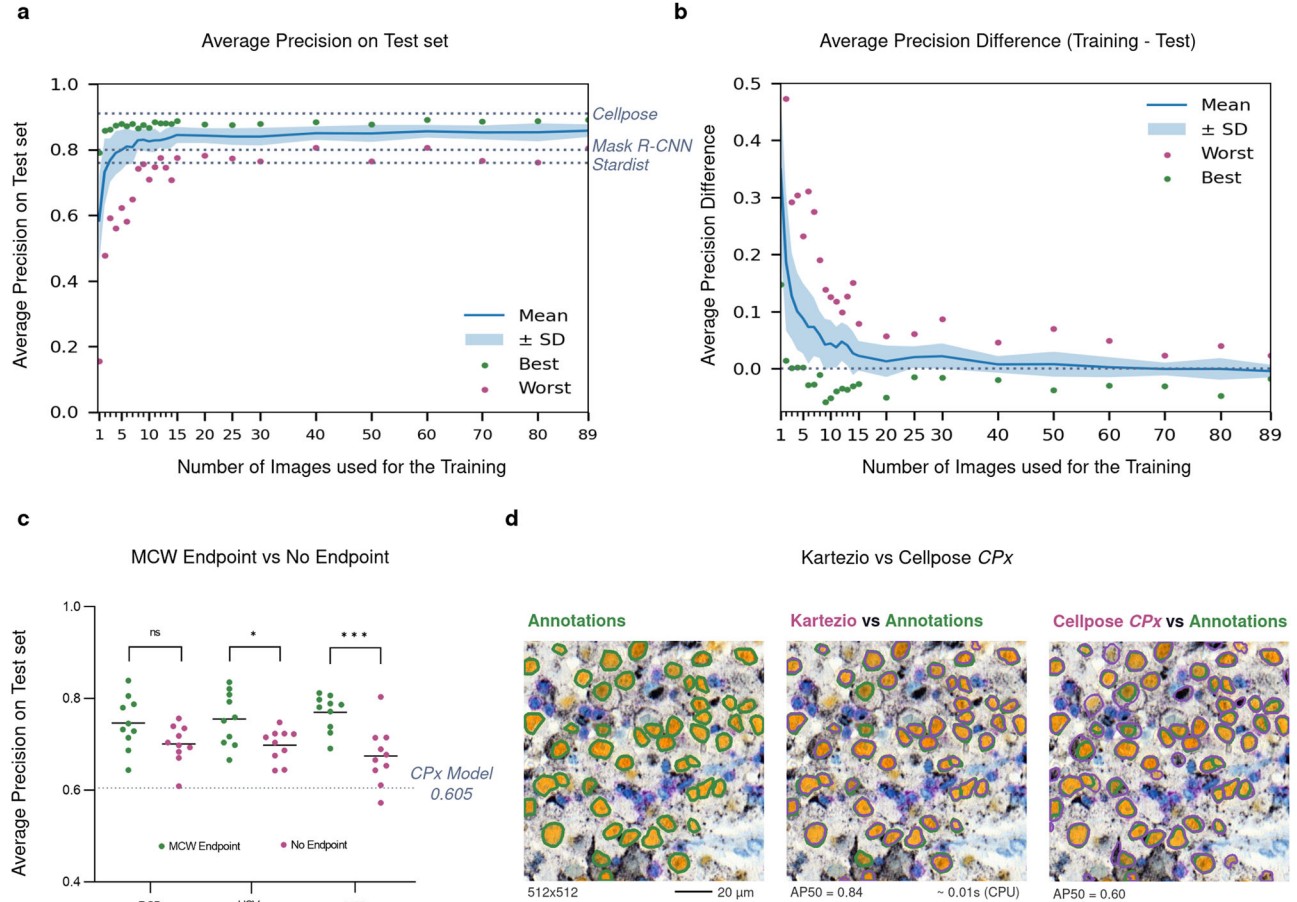

**Fig. 2 | Kartezio-generated models achieve comparable accuracy as state-of-the-art Deep Learning approaches with fewer training requirements.**

**a**, **b** Kartezio was trained on 1 to 89 images, derived from Cell Image Library[23,28]. Original images available from: Weimiao Yu, Lee H.K., Hariharan, S., Bu W.Y., Ahmed, S. CIL_40217, Mus musculus, Neuroblastoma. http://cellimagelibrary.org/images/40217. (2012). Its performance on a test set of 11 images was compared to Cellpose, Mask-RCNN and StarDist (each trained on 89 images; dotted lines).
**a** Average Precision (AP50) scores are shown for the indicated number of training images (mean +/- SD of $n = 35$ models per condition; each model represents an independent experiment). **b** Average differences in score between training and test datasets (mean +/- SD of $n = 35$ models per condition; each model represents an independent experiment). Dotted line indicates optimal difference of 0. Dots indicate best (*green*) and worst (*red*) scores obtained by Kartezio for the indicated number of training images. **c**, **d** For each set of conditions, ten Kartezio models were trained on twelve images from the melanoma cohort[29]. Each model is indicated by one data point; $n = 10$ independent experiments per condition. In parallel with the best generalist Cellpose model (CPx), these models were evaluated on a

test dataset of ten images. **c** AP50 scores for models trained using no defined Endpoint versus models trained using Marker Controlled Watershed (MCW) endpoint combined with different preprocessing methods (RGB, an HSV transformation, and a HED deconvolution) for ten Kartezio-generated models and compared to the AP50 of CPx. Each datapoint represents a single Kartezio-generated model ($n = 10$ independent experiments per condition). Bars indicate mean of ten experiments. Dotted line indicates score of CPx (0.605). MCW versus the No Endpoint control was compared using unpaired 1-way ANOVA with multiple comparisons. **d** Representative histology image from the test dataset with Instance Segmentation (IS) performed by Kartezio or Cellpose, compared to manual annotations. Left: manual annotations (green outline); middle: IS performed by the Kartezio-generated model represented in Fig. 1 (*magenta* outline) and compared to ground truth (*green* outline); right: IS performed by CPx (*magenta* outline) and compared to ground truth. (*$p < 0.05$, ***$p < 0.001$, ns = non-significant). Source data for all experiments are provided with this manuscript and summary statistics are listed in Supplementary Table 2.

$t = 7.236$; $HSV_{MCW}$ $p = 0.000018$, $t = 8.196$; $HSV_{No\ Endpoint}$ $p = 0.000015$, $t = 8.369$; $HED_{MCW}$ $p = 0.000003$, $t = 13.30$; $HED_{No\ Endpoint}$ $p = 0.00695$, $t = 3.479$, by one-sample $t$-test). Furthermore, the addition of the MCW endpoint significantly improved Kartezio's performance ($F = 5.649$; $p = 0.0003$, 1-way ANOVA) compared to the No Endpoint control, when used in combination with either HSV ($p = 0.0449$) or HED ($p = 0.0003$) preprocessing (Šidák's multiple comparisons test). To visualize the differences between Kartezio and Cellpose, annotations of the ground truth (manual annotation, left) and the segmentation predicted by each model (middle, right) for a representative image from the test set are provided in Fig. 2d. In the middle and right panels, nuclei that have been correctly identified by the models are circumscribed with similar outlines in both green (manual annotation) and magenta (model annotations); nuclei that failed to be identified by the model (false negative) appear in green only, while areas that have been

incorrectly identified as nuclei (false positive) are circumscribed in magenta only. In this image, Kartezio achieved an AP50 of 0.84 while Cellpose achieved an AP50 of 0.60; however, a modest number of errors such as those mentioned above are visually detectable. Of note, the model of Kartezio used to illustrate this example is represented in Fig. 1 and its corresponding generated Python class is shown in Supplementary Fig. 3. Thus, Kartezio is capable of generating a highly effective IS pipeline, containing just four common image-processing functions preceding the MCW, which is fully explainable to humans and implemented using only a few basic lines of code (Supplementary Fig. 3). The execution time of the whole pipeline for a 512×512 pixel image such as that shown in Fig. 2d is a mere 0.01 s, using one CPU of one laptop.

In the following sections of this article, we highlight the versatility of Kartezio. We first demonstrate the performance of Kartezio in the SS

of patient tissue sample images based on a limited number of annotations (Use Case 1), underlining the complete explicability and accessibility of the generated models. Both explicability and accessibility are key features in clinical settings. We then illustrate three additional Use Cases in which Kartezio successfully performs IS of biological entities (from nanoscale extracellular particles to microscale immune cells), again based on very few annotations. All the images that we included in our study are conceptually focused on the theme of cytotoxic T lymphocyte (CTL) and tumor cell interactions in the context of tumor immunology.

Supplementary Table 3 outlines the key features of dataset preparation and analysis for the four different Use Cases in the portfolio. Supplementary Table 4 summarizes the model parameters for each Use Case and Supplementary Table 5 includes the fitness scores (min, max, and mean) for testing and training datasets. All four Use Cases utilize the same default library of 42 functions (Supplementary Table 1) and the same hyperparameters (e.g. number of functional nodes, mutation rates, etc.). Of note, the library and hyperparameters can be still adapted by the user if requested. We have also included examples of possible post-segmentation analyses for each Use Case (performed after image segmentation by Kartezio and listed in Supplementary Table 3); these are for illustrative purposes and can be modified to answer different experimental questions without changing the function of Kartezio.

## Use Case 1: Explainable and accessible models for semantic segmentation of tumor nodules in patient tissue samples

We propose an end-to-end translational application of Kartezio for tumor nodule SS in patient tissue samples. The goal is to provide clinicians with an easy-to-apply and fully interpretable tool that is automatically generated starting from a small annotated dataset. For an initial application, we profited from an off-the-shelf tissue sample cohort of metastatic melanoma patients we have previously described[29]. In this application, as shown in Fig. 3a, Kartezio was employed to create a SS mask of tumor nodules within IHC-stained tissue samples. Melanoma cells and CTLs were identified by Sox10 (a nuclear marker of melanocytic lineage, orange) and CD8 (a CTL marker, purple) expression respectively and cell lysosomal content was identified by CD107a staining (black). All cells were counterstained with hematoxylin (blue).

In the present study, images were preprocessed by transforming RGB channels into HSV color space (Supplementary Fig. 2c). We limited the analysis to a dataset containing 24 images. Supplementary Fig. 4 depicts the 12 images used for training (Supplementary Fig. 4a) and the 12 used for testing (Supplementary Fig. 4b). Both image series have been annotated by an expert pathologist (green lines); illustrative examples are shown in Fig. 3a. Based on 100 runs, Kartezio obtained a mean IoU score of 0.87 ($\pm$0.018) on the training dataset, with a maximum score of 0.91, as shown in Supplementary Table 5. Kartezio obtained a mean score of 0.85 ($\pm$0.047) on the test dataset, with a maximum score of 0.89.

For each image in the training and test dataset, we merged the predictions of $n = 100$ trained models to generate composite heatmaps providing integrated representations of the obtained masks, as shown in Fig. 3a and Supplementary Fig. 4. These composite masks can also be evaluated against the ground truth to provide IoU scores. As shown in Supplementary Fig. 4c-d, we evaluated the fitness of the ensemble of 100 pipelines at multiple different threshold values (between $t = 0.0$ and $t = 1.0$ in increments of 0.01; illustrative images shown at $t = 0$, 0.35, 0.5, 0.6 and 0.75). This analysis demonstrated that fitness is highly dependent upon threshold value; using the training dataset, we calculated an optimal threshold of $t = 0.35$. Across the 12 tumor nodules of the test dataset, the pipeline ensemble yielded a mean fitness score of 0.90. At this optimal threshold, the ensemble outperformed individual pipelines on average and even outperformed

the best individual pipeline, which had a score of 0.89 on the test set, illustrating the robustness of the ensemble approach for semantic segmentation. It is worth noting that fitness scores ranged from 0.81 to 0.96 across the 12 images for the test dataset, indicating that the pipeline ensemble performed consistently (with an SD between images of 0.054) and with high degree of accuracy in every tumor nodule tested, an important feature for clinical applications.

Notably, each of the 100 Kartezio-generated algorithms can be interpreted and explained. As an example, we selected the pipeline that obtained the best IoU score and dissected its behavior (Fig. 3b). The first step (node 3) of this pipeline consisted of subtracting the Hue channel (node 0) from the Saturation channel (node 1), allowing for the exclusion of pixels corresponding to CD8$^+$ T cells and to other cells belonging to the tumor microenvironment. Then Kartezio used a Gaussian Blur filter (node 4) in order to smooth the previously obtained area. Next, Kartezio chose a Laplacian filter (node 10) to detect the variations in the image obtained in node 4, thus delimiting the tumor borders. Interestingly, the sequence Gaussian Blur/Laplacian filters is frequently used by image analysts for image segmentation tasks. To highlight the segmented areas, Kartezio used Morphological Black Hat (node 17) which reveals small objects that are darker than their surroundings. Altogether, these initial calculations provided an accurate estimation of tumor area.

In parallel, node 12 created a binary mask of the Value channel (brightness) allowing for the establishment of a global map of the area of interest. The next step selected by Kartezio consisted of merging the two maps with the min operator (node 19). This allowed for the elimination of all pixels detected by the first calculation (results of node 17) that fell outside the area of interest. Kartezio next selected a Fill Hole filter in order to transform the selected area limits into a solid binary mask. A final step consisted of removing the noise from the edges of the mask using a Median Blur filter (node 31).

In its entirety, the described pipeline allowed for the sharp delimitation of tumor nodules with a high degree of accuracy. Crucially, a step-by-step dissection of Kartezio's pipeline, similar to the one we outlined above as an example, can be applied to each pipeline.

We next wanted to determine whether the model ensemble that generated the initial 100 masks on the low resolution IHC images was capable of demarcating tumors in a higher resolution context, using the optimal threshold ($t = 0.35$) calculated in Supplementary Fig. 4c. As shown in Fig. 3c, d, a technically challenging segmentation area of the tumor nodule pictured in Example 1, Fig. 3a (see purple box inset) was used to test this concept using two different strategies. First, as shown in Fig. 3c, the heatmap previously generated from 100 masks on low resolution training level images was upsampled and applied to high resolution images (Level 7 and 5). Second, as shown in Fig. 3d, the ensemble of 100 pipelines derived from the low resolution training set was deployed on higher resolution images, generating 100 new masks at each level, which were subsequently merged into a single heatmap as shown. Remarkably, in this illustrative example, the heatmap generated by this second strategy demarcated the tumor boundaries in close accordance with the high resolution IHC images, despite the region being technically difficult to segment. Furthermore, neither of these two strategies involved re-training Kartezio on high resolution images, but rather repurposing the masks (first strategy) or the pipeline ensemble (second strategy) from the low resolution dataset. The ability to train Kartezio on low resolution images to generate pipelines that perform well on high resolution images provides a substantial computational advantage, since the extremely cumbersome amount of data associated with high resolution IHC images makes them challenging to use in a training context.

This easily interpretable tool to rapidly segment tumor nodules has the potential to allow automatic assessment of various parameters that are critical for cancer patient management, such as tumor nodule size, morphological features, tumor localization, its borders with

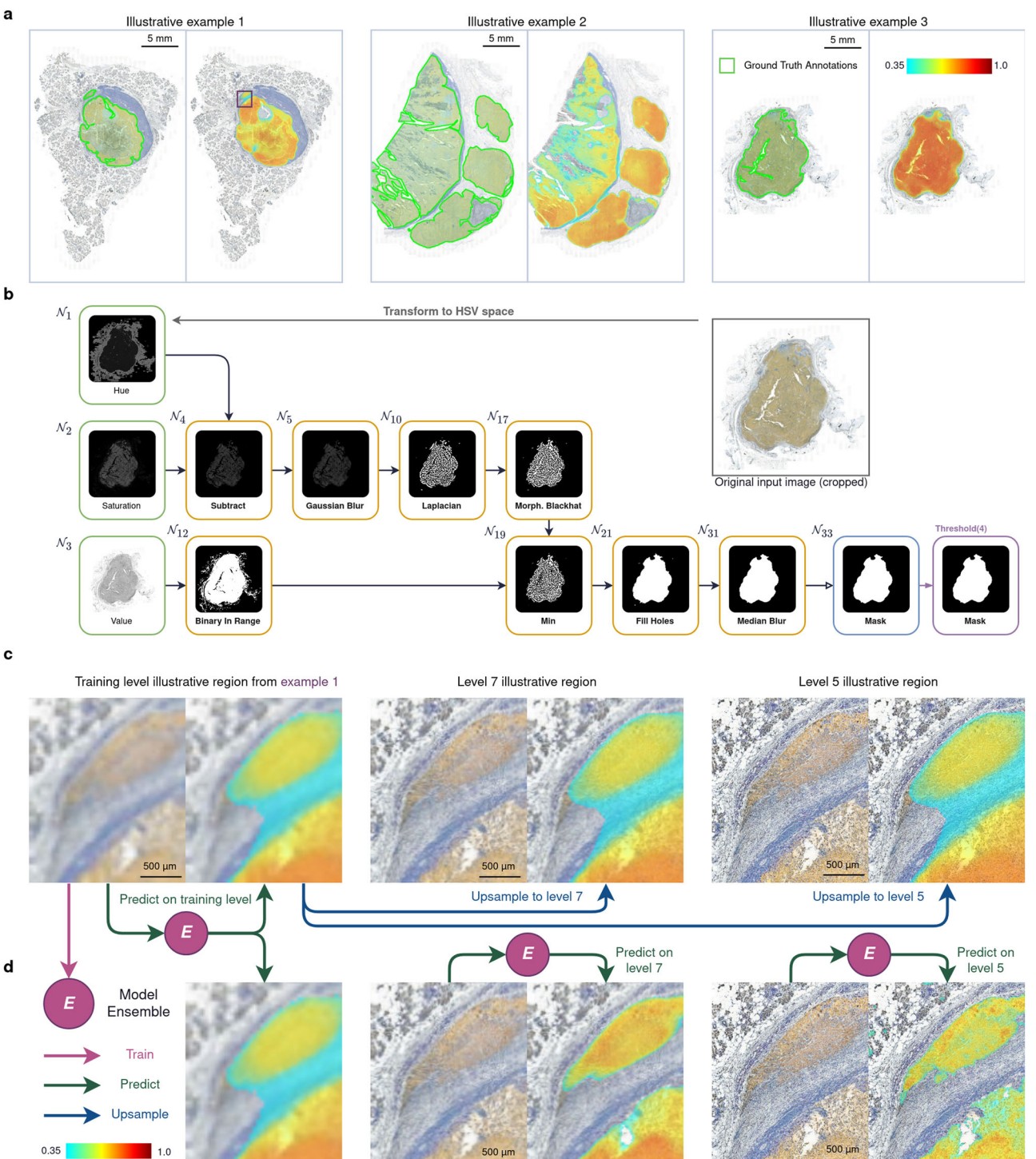

**Fig. 3 | Segmentation of tumor nodules within IHC-stained tissue specimens.**
*Use Case 1:* **a** Tissue slices stained for melanoma marker Sox10 (*orange*), CTL marker
CD8 (*purple*), lysosomal marker CD107a (*black*) and hematoxylin (*blue*) were used
for training and testing of Kartezio-generated semantic segmentation pipelines
(three illustrative tumors shown; full dataset in Supplementary Fig. 4). Left panels
depict tumor contours delineated by an expert pathologist (*green* lines). Right
panels depict predictions of an ensemble of $n = 100$ Kartezio-generated pipelines,
with outputs averaged to generate composite heatmaps of predicted tumor con-
tours using an optimized threshold value of $t = 0.35$ (i.e. minimum probability
depicted is 0.35; maximum is 1.0). **b** The pipeline with the best IoU score (out of
$n = 100$ independent experiments) is shown. The mask generated by this pipeline
was combined with masks predicted by 99 other pipelines to create the $n = 100$

model ensemble (indicated by E), which formed the basis of heatmaps shown in (**a**).
**c**, **d** Two strategies were tested for upscaling existing pipelines to high resolution
images using an illustrative region (*purple* box inset) of the tumor nodule in (**a**).
**c** The heatmap generated from the model ensemble trained on low resolution
("training level") images was upsampled and overlaid onto the high resolution
images (Level 7 and Level 5). **d** The model ensemble trained on training level images
($n = 100$) was deployed on high resolution Level 7 and Level 5 images to generate
100 de novo masks on each level, which were merged to generate a new heatmap
corresponding to each level ($t = 0.35$; minimum probability depicted is 0.35; max-
imum is 1.0). Source data for all experiments are provided with this manuscript and
summary statistics are provided in Supplementary Table 5.

surrounding tissue, and distance from the resection margins, as described in more detail in the discussion. In the next sections, we illustrate three additional Use Cases in which Kartezio successfully performs IS in biological images of different scale and resolution.

## Use Case 2: Multi-molecular scale: detecting and measuring CTL-derived extracellular particles

Recent studies revealed that the fight between CTLs and tumor target cells is not limited to lytic synapses (specialized cell-cell contact areas where cytotoxic lytic granules are secreted from killer CTLs[30,31]) but also occurs via the release of extracellular aggregates of lytic components and additional bioactive molecules called Supra Molecular Attack Particles (SMAPs)[32,33]. SMAPs are currently under investigation as potential anti-cancer therapeutics and may represent a potent new tool in the arsenal of oncologists. It is therefore crucial to develop new methods to characterize these particles in order to better define their biological function and therapeutic potential. As individual CTL-derived SMAPs are small (~120 nm in diameter, as detected by D-STORM imaging)[33], high resolution imaging techniques are required for their visualization.

In a first application of Kartezio, we acquired high resolution Total Internal Reflection Fluorescence Microscopy (TIRFM) images of particles released from human polyclonal CD8+ CTLs (derived from two donors) and stained with Alexa 647-conjugated wheat germ agglutinin (WGA; a red fluorescent probe that stains sialic acid and N-acetyl-glucosamine moieties of glycoproteins), and DiO (a green fluorescent lipophilic dye that stains hydrophobic lipid membranes); individual channels from a representative image are displayed in Supplementary Fig. 2d. The population of particles included WGA+ (cyan), DiO+ (magenta) and double positive particles (lavender) (Fig. 4a). The aim of this test paradigm was to determine if Kartezio could distinguish putative SMAP particles (WGA+DiO-) from a mixed population of extracellular particles and vesicles, and then to extract quantitative information about the different particle populations. Although conceptually simple, this Use Case enabled us to demonstrate the performance of Kartezio in the context of single-particle high resolution imaging.

Using Kartezio, we created an automated procedure to perform IS, which permitted the subsequent identification of single- and double-positive particles and characterization of their features. For each channel, we first trained Kartezio on a dataset consisting of one manually-labelled image, split into quarters (the number of ROIs is indicated in Supplementary Table 3). Based on 35 training runs, Kartezio achieved an average AP50 score of 0.81 (±0.037) and 0.79 (±0.030) for the WGA and DiO channels respectively, with a maximum score of 0.87 and 0.84 for WGA and DiO respectively. To assess generalizability, all IS pipelines were then tested on a dataset of one manually-labelled previously unseen image, split into quarters. Under these test conditions, Kartezio achieved an average AP50 score of 0.73 (±0.083) and 0.76 (±0.090) for the WGA and DiO channels respectively, with a maximum score of 0.84 for WGA and 0.85 for DiO. These results are summarized in Supplementary Table 5.

For each channel, the best overall pipeline (based on test and training fitness scores) was then applied to an image cohort containing an additional 9 images of CTL-derived extracellular particles. For the WGA channel, the best overall pipeline had a training fitness score of 0.87 and a test fitness score of 0.84. For the DiO channel, the best overall pipeline had a training fitness score of 0.81 and a test fitness score of 0.85. Figure 4b shows a representative example of cyan and magenta particle segmentation obtained with the best pipeline for each channel out of the 35 runs performed. In order to identify double-stained particles post-segmentation, the WGA and DiO instances previously detected using Kartezio were tested for matching using a pairing mechanism based on the Intersection over Union (IoU) metric (detailed in Methods). Instances detected in the WGA and DiO channels with the IoU score above a threshold of 0.05 were identified as double-positive instances. The entire process enabled the identification of three sets of particles: WGA+DiO- (cyan), WGA-DiO+ (magenta), and WGA+DiO+ (orange) (Fig. 4c).

From these datasets we could readily calculate the mean fluorescence intensity (MFI) and area of each instance as shown in Fig. 4d. The panel displays a scatterplot of WGA+DiO- (cyan), WGA-DiO+(magenta), and WGA+DiO+ particles (orange) released from CTLs. This analysis proved capable of distinguishing extracellular vesicles (WGA-DiO+ and WGA+DiO+ particles) from putative SMAPs, which were previously shown to be WGA+ but negative for lipophilic plasma membrane dyes[33].

Taken together, the above results show that Kartezio can successfully perform rapid and accurate IS of extracellular particles at a multi-molecular scale, providing access to further analysis of a

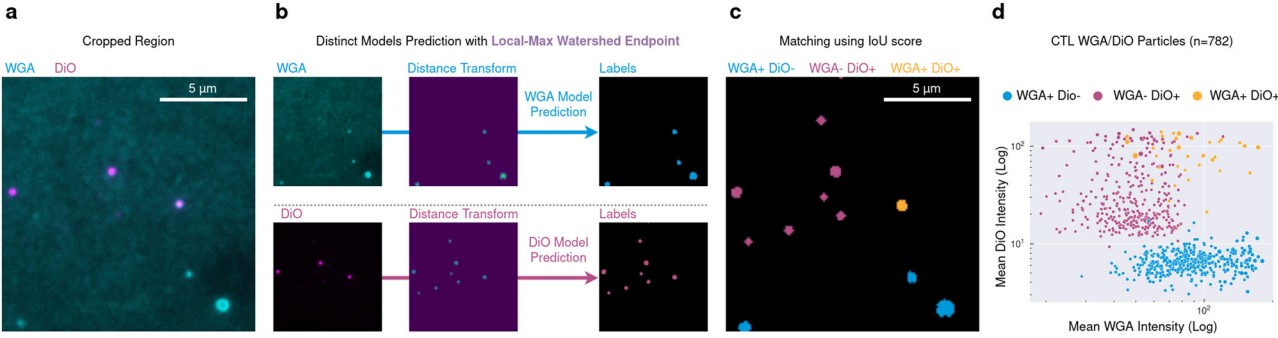

**Fig. 4 | Analysis of extracellular particles using Kartezio-mediated instance segmentation.** *Use Case 2:* CTL-derived extracellular particles stained with WGA and DiO were imaged using TIRFM (see Supplementary Fig. 2d). Total Internal Reflection Fluorescence Microscopy (TIRFM) images of particles released from human polyclonal CD8+ CTLs (derived from two donors) and stained with Alexa 647-conjugated wheat germ agglutinin (WGA; a red fluorescent probe that stains sialic acid and N-acetyl-glucosamine moieties of glycoproteins), and DiO (a green fluorescent lipophilic dye that stains hydrophobic lipid membranes). **a** A typical cropped image depicting extracellular particles that were positive for either WGA (*cyan*), DiO (*magenta*) or both. **b** The two images on the left depict the individual channels (WGA and DiO). For each channel, Kartezio generated a model for instance segmentation (arrows) leading to labels via the Local Max Watershed

Endpoint. Each label corresponded to one instance (particle) in a given channel as shown. **c** To begin the post-segmentation analysis, each label was classified as WGA+ (*cyan*), DiO+ (*magenta*) or WGA+DiO+ (*gold*), with matching evaluated on the basis of label overlap using the Intersection over Union (IoU) metric. **d** Feature extraction from each instance involved quantification of particle size and mean fluorescence intensity (MFI). Individual dots represent WGA+ (*cyan*), DiO+ (*magenta*) and WGA+DiO+ double positive particles (*gold*) plotted on the basis of MFI. Dot size is proportional to the size of the particle mask corresponding to each instance. Representative data shown are derived from one experiment (i.e. one model per channel) from a total of n = 35 generated models per channel. Source data from n = 35 experiments are provided with this manuscript and summary statistics are provided in Supplementary Table 5.

statistically relevant number of events. It is important to note that the pipelines generated by a Kartezio can be exported to standard Python scripts, which can be readily re-used on additional images acquired in comparable experimental conditions.

## Use Case 3: Sub-cellular scale: assessing the lytic arsenal of individual CTLs

To test Kartezio's performance on a larger scale, we assessed the capacity of Kartezio to automatically measure expression of various molecules within cells, in order to classify individual cells based on their molecular content. Here we focused on the CTL lytic arsenal by measuring the expression of various lytic granule components (e.g. perforin, granzyme B, and CD107a) in polyclonal human CTLs. Cells were stained with antibodies directed against CD45 (grayscale), perforin (magenta), granzyme B (yellow) and CD107a (cyan) (see Supplementary Fig. 2e). Staining for CD45, a molecule expressed on the surface of all hematopoietic lineage cells including CTLs, allowed Kartezio to delineate individual cells, while the other markers were subsequently used to score the CTL lytic arsenal during the post-segmentation analysis.

For the initial analysis, we applied Kartezio to series of z-stacks containing CD45 staining only (Fig. 5a). To achieve this aim, we upgraded Kartezio to segment cells in 3D stacks to provide accurate 2D masks of each identified cell. Kartezio-generated pipelines were sequentially applied to each layer of the z-stack to produce initial Watershed masks of each z-stack. All the masks were merged with an averaging operator which allowed us to obtain the maximal area of the cells in the z-dimension (Fig. 5b). Local Max Watershed was applied to produce the IS masks of the cells (individual segments, Fig. 5c).

For this analysis, the training fitness was more stringent than in the previous analysis (AP with a threshold of 0.7 instead of 0.5) to avoid

overfitting. The training dataset was limited to 5 images containing 45 cells, highlighting the aforementioned capacity of Kartezio to be trained for cell segmentation on a very small set of images. On 35 independent runs, Kartezio obtained an average score of 0.98 ($\pm 0.007$) on the training dataset ($n = 5$ images with 45 cells) and 0.83 ($\pm 0.028$) on the test dataset ($n = 4$ images containing 34 cells), with a maximum score of 1.000 during training and 0.89 during testing (results shown in Supplementary Table 5). The best overall pipeline had a score of 1.00 during training and 0.85 during testing.

After the individual cells were segmented using Kartezio's best pipeline, we measured staining of the three lytic granule components (perforin, granzyme B, and CD107a) in each z-section to derive a single feature vector for each cell. This vector was a composite of different quantitative readouts and integrated both individual and combined counts of positive pixels, sums of pixel fluorescence intensities, and average intensities measured in the different channels (see Material and Methods). Each cell was then positioned on the UMAP[34] based upon this composite feature vector. The gradient of expression of each lytic granule component within cells of the whole population is shown in Fig. 5d–f. Figure 5g shows the gradient of expression in the whole population for the three markers together.

Together these results underline the capacity of Kartezio to serve as a versatile tool for the analysis of molecular content in cells. Importantly, this function can be used more broadly to automatically assess staining intensities in a wide range of microscopy images, one of the most common tasks in image analysis in cell biology.

## Use Case 4: Cell-cell interface scale: discerning CTL/target cell lytic synapses within a mixed population of cells

In the next step of Kartezio's validation, we focused our analysis on lytic synapses, which are highly ordered structures that form when a

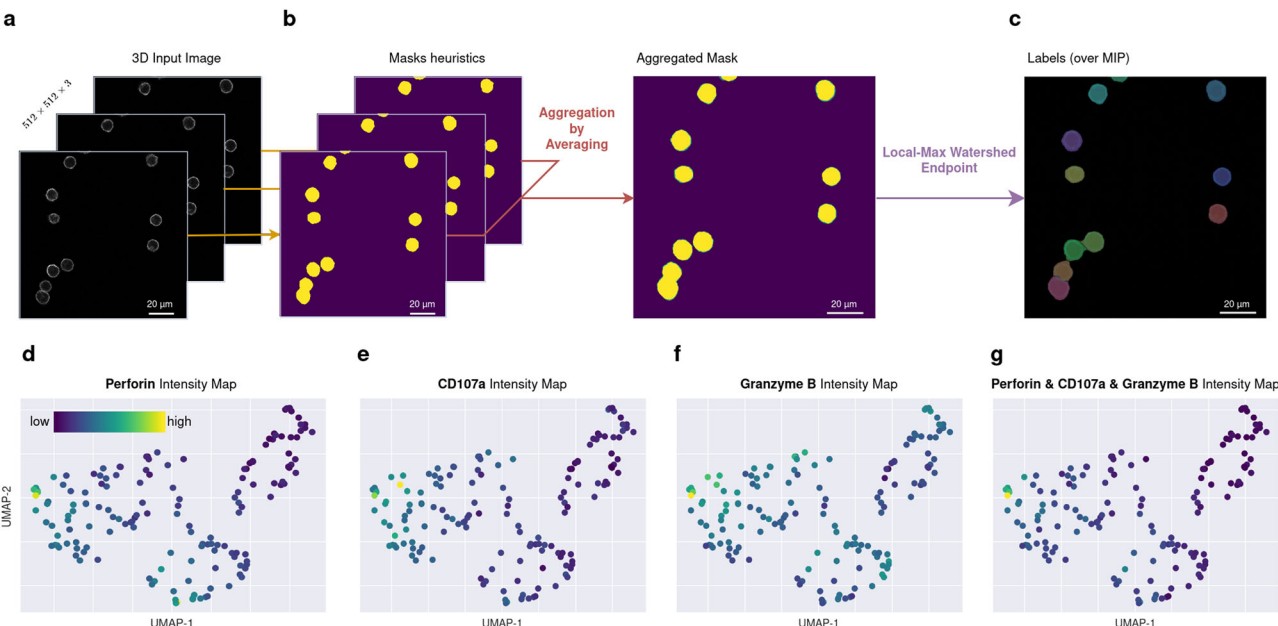

**Fig. 5 | 3D analysis of lytic molecule content in individual CTLs within a polyclonal population.** *Use Case 3:* Human polyclonal CD8⁺ T cells were stained with antibodies directed against CD45, perforin, granzyme B and CD107a (see Supplementary Fig. 2e). **a** z-stacks of images in the CD45 channel were used as a 3D input to delimitate individual cells. **b** Kartezio generated an instance segmentation pipeline that was sequentially applied to each layer of the z-stack to generate the corresponding mask. The masks were aggregated using an averaging operator. **c** The application of the Local Max Watershed Endpoint allowed the generation of 2D labels. **d**–**f** For the post-segmentation analysis, the MFI of perforin, granzyme B, and CD107a was calculated for each instance to derive a single feature vector for

each cell. This vector was a composite of different quantitative readouts and integrated both individual and combined counts of positive pixels, sums of pixel fluorescence intensities, and average intensities measured in the different channels. Each cell was visualized on a 2D UMAP representation[34] based upon this composite feature vector. **g** The same UMAP representation of merged lytic components. Representative data shown are derived from one experiment, from a total of $n = 35$ independent experiments. Source data for $n = 35$ experiments are provided with this manuscript and summary statistics are provided in Supplementary Table 5.

CTL productively engages a target cell that expresses its cognate antigen[30,31]. A CTL/target cell interaction that forms around a lytic synapse is known as a cell-cell "conjugate". Identifying a CTL/target cell conjugate is a complex task, which involves correctly identifying and demarcating different cell types and assessing whether two adjacent cells are interacting. Herein, our objective was to design an automatic procedure which would take as input 3D stacks of images of CTLs interacting with target cells to identify the CTL/target cell conjugates.

To achieve this, cells were stained with DAPI (cyan, to detect CTL and target cell nuclei), with antibodies directed against a-tubulin (green, to visualize the cytoskeleton in both cells), and perforin (magenta, a lytic molecule expressed only by CTLs) to allow Kartezio to distinguish the CTLs from the targets (Fig. 6a); individual channels from a representative image are displayed in Supplementary Fig. 2f.

As previously illustrated in Fig. 5b, Kartezio-generated pipelines were sequentially applied to each layer of the z-stack. This process generated paired images of masks and markers (Fig. 6b). All the obtained pairs were then merged using an averaging function (see Methods) to produce one mask heatmap and one marker heatmap (Fig. 6c). Watershed was then applied to the heatmaps to produce 2D masks containing distinct segments of cells (Fig. 6d). At this step, Kartezio achieved an average AP50 score of $0.75 \pm 0.022$ over 35 runs during training with a maximum score of 0.81, and an average score of $0.67 (\pm 0.034)$ during testing on a validation set containing 150 previously unseen cells. These results are summarized in Supplementary Table 5. The best overall pipeline had a training score of 0.79 and a testing score of 0.72, and was utilized for the subsequent post-segmentation analysis.

At this stage of the image analysis, the segmented instances generated by Kartezio (each corresponding to exactly one cell, either CTL or target cell) were further analyzed using conventional image analysis approaches. Knowing the average dimensions of CTLs and target cells, we excluded all entities with a surface <350 pixels ($\sim 24 \mu m^2$) and > 6000 pixel ($\sim 417 \mu m^2$), which might correspond to cell fragments or big cellular clusters. Additionally, we also applied a morphological closing to smooth cell contours. The next step, depicted in Fig. 6e, was to classify each instance as either CTL or target cell. To this end, we calculated the area and the intensity of DAPI (cyan), a-tubulin (green) and perforin (magenta) fluorescence for each instance. We considered these four features sufficient to separate the two cell populations since CTLs are statistically smaller than target cells and express perforin. Unsupervised ML using Principal Component Analysis (PCA)[35,36] followed by Gaussian Mixture Model (GMM, see Material and Methods) was then used to separate the two

population of cells. This method allowed us to reach an accuracy of 92.2 % on a validation set containing 153 cells, as evaluated against the ground truth established by independent analyzers.

Once the CTL and target cells were identified, we selected CTL/target cell conjugates based on their contiguity using the following two complementary procedures: i) we calculated the distance between the centroid of the CTL and the target cell masks and kept only the cells with a distance less than 70 pixels ($\sim 18.5 \mu m$). This procedure allowed us to keep in the analysis CTL and target cells with high probability of contact; and ii) in the selected conjugates, each target cell mask was slightly dilated to potentially intersect with other masks. This procedure was required since IS produces distinct (i.e. non-overlapping) masks. If an intersection with a CTL mask was detected, the target cell was considered in conjugation with the T cell (Fig. 6f).

Cumulatively, our procedure allowed us to keep in the analysis bona fide conjugates while excluding big cellular clusters and cell fragments. Each conjugate included in the analysis was verified by independent analyzers, showing an accuracy of the combined procedure of 75,4% on 52 lytic synapses.

In conclusion, the above results highlight a two-step approach, in which Kartezio can identify cells that are then classified by ML into different cell populations on the basis of their size and staining features. This offers the opportunity to explore automatic high-throughput scoring of defined parameters of immunological synapse formation and of other cell activation/death parameters in cell-cell conjugates. We show that Kartezio can synergize with state-of-the-art ML methods to create a hybrid AI approach capable of assessing and interpreting complex functional structures in biological images.

## Discussion

In the present study, we release an innovative new computational tool called Kartezio for image segmentation of biological and medical images that draws inspiration from Darwinian evolution to automatically generate image processing pipelines. When compared head-to-head with DL approaches, Kartezio consistently achieved scores that were competitive with state-of-the-art approaches (Fig. 2, Supplementary Table 2), despite being trained on a very limited number of input images. We applied our approach to four distinct Use Cases (Figs. 3–6, summarized in Supplementary Table 3) related to tumor immunotherapy, a highly dynamic research area in contemporary medicine wherein automated image analysis has garnered significant interest[37,38]. We validated Kartezio on tasks ranging from the characterization of nanometer-scale extracellular vesicles released by CTLs (average mask diameter of ~0.6µm, based on manual user annotations)

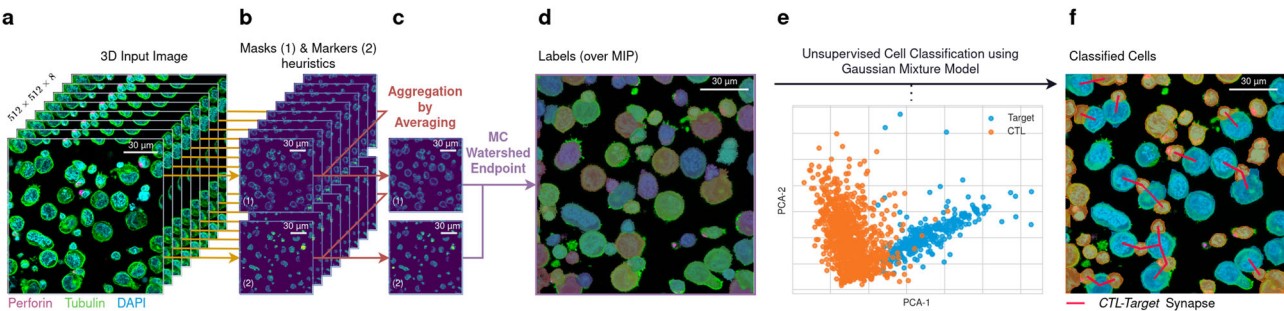

**Fig. 6 | A hybrid Kartezio/AI approach to detect CTL/target cell conjugates.** *Use Case 4:* (**a**) A mixed population of interacting cells containing both human clonal CTLs and target cells was stained with DAPI (*cyan*) and antibodies directed against α-tubulin (*green*) and perforin (*magenta*), as shown in Supplementary Fig. 2f. Each series of z-sections was used as a 3D input. **b** The same Kartezio-generated instance segmentation pipeline was applied to each z-section to generate pairs of masks and markers. **c** All masks and markers were merged to obtain one 2D mask heatmap and one 2D marker heatmap. **d** The Watershed Endpoint was applied to generate 2D

labels. **e** During post-segmentation analysis, unsupervised Machine Learning (Principal Components Analysis followed by Gaussian Mixture Model) was used to classify the two-cell population. **f** Visualization of cell classification. Colored masks indicate the two classes of cells, CTLs (*orange*) and target cells (*blue*). Red lines indicate putative synapses between CTLs and target cells. Representative data shown are derived from one experiment, from a total of $n = 35$ independent experiments. Source data are provided with this manuscript and summary statistics are provided in Supplementary Table 5.

to micrometer-scale analysis of CTL phenotype (average mask diameter of ~12μm, based on manual user annotations), and ultimately to centimeter-scale detection of cancer nodules in patient tissue samples (average estimated mask diameter of ~1.1 cm, based on pathologist annotations). This stringent testing strategy provided a crucial validation of Kartezio's capabilities across a wide variety of image types and across scales differing by five orders of magnitude.

A conceptual step-by-step summary of the Kartezio workflow is illustrated in Supplementary Fig. 1. The experimentalist first prepares the training and testing datasets by manually annotating a limited number of images. Then, the training images are provided to Kartezio, which generates initial pipelines composed of functions randomly drawn from the default function library, randomly parameterized and randomly arranged into an image processing pipeline. Each pipeline is evaluated against user defined annotations and the best pipeline is selected to proceed through the evolutionary selection process (illustrated in Supplementary Movie 1 using images from the Cellpose dataset, originally derived from Cell Image Library[23,28]). The optimized pipeline proceeds to the testing phase and, if satisfactory, it can be then deployed on a previously unseen dataset.

As demonstrated in the four Use Cases, once image segmentation (IS or SS) has been achieved using Kartezio, post-segmentation analysis of the segmented images can be performed, ranging from simple feature extraction from instances (e.g. MFI, size of particle, etc.) to ML-based analyses, depending upon the experimental design and needs of the experimentalist (illustrated for the four Use Cases in Supplementary Table 3).

Our work tackles two major scientific challenges in the fields of CV and biomedicine. Firstly, it is becoming increasingly important to generate fully explicable image analysis strategies and to track the logic of decisions made by algorithms to ensure responsible patient management. Unlike Kartezio, the so-called "black box" DNN networks are difficult for humans to analyze and interpret, in spite of recent progress in the field of XAI[5,39]. Limited interpretability is a potentially important drawback, particularly in the clinical context, where the process of extracting information from medical images should be fully transparent and interpretable in order to foster trust, ensure compliance with professional ethical standards, and facilitate regulatory oversight. This is also in line with the main principles of the European Union AI Act[40], currently under discussion by the European Council, that regulates the use of AI in sensitive fields such transportation and health care. As shown in Figs. 1 and 3, as well as Supplementary Fig. 3, Kartezio is natively interpretable, meaning that the generated pipelines can be humanly read, evaluated and tested for provability. Given the potential applications for Kartezio in clinical pathology and related fields, its fully interpretable "white-box" nature presents a major conceptual advantage[14].

Recent advances in Large Language Models (LLM) are also in line with the quest for more interpretable data analysis pipelines. Indeed, LLM such as ChatGPT[41,42], SantaCoder[43] and GitHub Copilot[44] can directly generate the code necessary to analyze a given image or dataset, provided that appropriate prompts are specified by the expert (e.g. type of input image, expected output, functions to use in the analysis pipeline, etc.). Further research is ongoing to identify the optimal strategy to generate such prompts, in order to guide LLM towards the generation of adequate models. Finally, LLM and GP are currently being used to generate ML pipelines capable solving complex tasks[45,46]. Each ML model could be trained to solve a specific and comprehensive task (e.g. extraction of tumor nodules, segmentation of cell nuclei, etc.) and the assembly of these individual ML models would build an interpretable pipeline at a higher scale. Interestingly, because of the low requirement of data and the simplicity to train the model, Kartezio could be used to produce these specific models before the pipeline is assembled at a higher level.

A second important challenge concerns image annotation, which is a time-consuming task that requires high biomedical expertise. It is therefore crucial to minimize experts' effort by designing ML algorithms that can handle the smallest datasets possible, while keeping adequate generalization capacity. To address this point, our study utilizes CGP to generate solutions that leverage the cumulative expertise of human CV experts. In other words, CGP produces pipelines by assembling algorithms that integrate decades of existing research and development in CV. While the number of images used in our study is already remarkably low, it is conceivable to upgrade Kartezio to an active "human-in-the-loop" training mode to further reduce the size requirement of the training dataset and improve performance. In such a configuration, intermediate results from Kartezio-generated pipelines could be displayed while the expert annotates. At a given point, the expert would not annotate anymore, but simply amend the automatically generated annotation. In turn, expert-operated adjustments would be informative to pursue the training of Kartezio.

In addition to these major conceptual advantages, Kartezio offers several concrete technical improvements over existing IS approaches. For example, Kartezio can handle images of diverse dimensions within the same dataset. In addition, it can be trained on small patches and later deployed on much larger images. As shown in Fig. 3, Kartezio can also be trained on low resolution images and the resultant pipelines applied to high resolution images, which removes the need to train the models on cumbersome high resolution images. Finally, Kartezio is not restricted to gray scale input images and can easily accommodate RGB, HSV, and HED formats.

Nonetheless, there are certain limitations to bear in mind when assessing the applications of Kartezio. For example, while the pipelines shown in Figs. 1 and 3 are clear, streamlined and explicable, there is no guarantee that all Kartezio-generated pipelines will be so intuitive to understand. This is particularly salient if the predetermined number of nodes is high and/or the number of available functions in the function library is high. In the field of evolutionary computation, research is ongoing to take into account the size of the graph in the evaluation process using multi-objective optimization[47–49]. Another complementary approach to control graph size can come from automatic modularization strategies of produced graphs[50]. Further research is required to explore and apply these solutions to Kartezio.

Our approach is particularly notable for its integration of human knowledge into the image analysis process. Expert knowledge is integrated into three specific areas: i) the construction of the function library (ii) the choice of hyperparameters, and (iii) the selection of non-evolvable components (in this case, preprocessing nodes and endpoints), discussed individually below.

Firstly, for the design of the function library, specific expertise is necessary to determine the minimal set of functions that can adequately solve the task while keeping the dimensions of the search space as small as possible. For the image processing applications in this manuscript, the function library represents a comprehensive expert-curated list of the most common image processing functions in the literature, and as such certain redundancies between similar common functions are not unexpected. The existing list was designed to maximize the diversity of functions available to Kartezio while ensuring that all key types of function were represented, including: morphological (e.g. Erosion, Dilation), arithmetic (e.g. Add, Sub, Square), logical (e.g. And, Or, Not), and edge detection (e.g. Laplacian, Sobel), amongst others. Ongoing syntactic graph studies are currently underway to determine the frequency with which individual functions are represented in final pipelines and the propensity of given function pairs (bigrams) or triplets (trigrams) to be used together (e.g. Gaussian Blur followed by Sobel Filter). Clusters of functions typically used together can be entered into the function library as new functions, and rarely used or redundant functions can be removed. These decisions can be made based upon real data derived from the analysis of large numbers of generated pipelines. Such data-driven insights will enable the function library to be fine-tuned, and expert users have the option to

further refine the function library, providing an additional opportunity for human knowledge to be integrated with the ML task. The current version of Kartezio contains 42 functions that have been manually selected to address the specific tasks of this study. A possible improvement of Kartezio would be to automatically select functions included in existing programming libraries to generate ad hoc CGP libraries suitable for a given task. In this case study, Kartezio was applied to the task of IS, but this approach could clearly be applied to other tasks in the CV field as long as pre-existing functions are available and already used by human-made algorithms.

Secondly, the selection of hyperparameters is an important consideration in the use of Kartezio. The hyperparameters currently used in this manuscript ($\eta = 30$ nodes, $\lambda = 5$ offspring over $\mathcal{K} = 20{,}000$ iterations) represent common hyperparameters in the CGP literature, and these could serve as default parameters or recommended settings for non-expert users of Kartezio. However, we envision that expert users may wish to modify certain hyperparameters to personalize the model to their needs and optimize performance, and further research is currently underway to assess the impact of alternative hyperparameters on generated pipelines.

Thirdly, the addition of non-evolvable nodes (such as preprocessing nodes or endpoints) is an important mechanism for integrating human knowledge into the image analysis pipeline, a feature that is also seen in contemporary DL approaches. In this study, we provide evidence (shown in Fig. 2c) that pipelines given a non-evolvable endpoint appropriate to the objective (in this case, Marker Controlled Watershed for IS) significantly outperform those with no endpoint indicated. We envision that non-evolvable endpoints should initially be selected by expert users based upon the specific objective of the image processing task (e.g. Hough Circle Transform for segmentation of round objects, Marker Controlled Watershed for IS), and these would become the suggested default settings for non-expert users, who could choose from recommended non-evolvable nodes based on indicated task objective. Likewise, as demonstrated in Fig. 2c, appropriate preprocessing nodes can be selected based on the properties of the input image, and their impact on model performance quantitatively assessed. Ultimately, automation of non-evolvable node selection may also be a powerful approach.

Although the design of the function library as well as the selection of hyperparameters and non-evolvable components are by nature subjective, the impact of these components on the performance of the model can be objectively assessed (as shown in Fig. 2c), which enables validation of the expert knowledge integrated into this approach.

It is important to note that in the present configuration, Kartezio only handles images. However, in biomedicine a plethora of data is collected in different formats and must be analyzed in an integrated fashion. In biology, these data may be derived from diverse molecular and functional experiments, as well as from microscopy. In medical practice, these clinical data may include laboratory test results, demographic, lifestyle and prognostic factors, and imaging data from diverse sources (e.g. histopathology, PET scans, etc). Mixed-Type CGP (MT-CGP)[51] is a promising approach to handle and integrate heterogenous data, such as scalars, vectors and matrices. Upgrading Kartezio in this direction would allow us to create data analysis pipelines that, while remaining fully explainable, would integrate data of different kind and origin. In clinical routine, such an approach would endow practitioners with AI-based clinical assistants that would be compatible with the upcoming regulatory guidelines.

In biological research, only a fraction of generated data is included in publications. Experimental data are indeed often analyzed only to answer specific questions, but are rarely used for broader integrated analyses. Such deep analyses are time and energy consuming and might not provide results immediately beneficial to the advancement of a given research. They may therefore be neglected. Kartezio could be trained at low human cost on small datasets to automatically extract predefined features (geometrical information from images, intensity of staining from flow cytometry data, activation of death pathways in target cells, single cell mRNAseq data, etc). These extracted data would feed large pre-analyzed data bases that could be used by data scientists to address new questions using mathematical and ML approaches.

An additional important benefit of Kartezio's application to microscopy concerns the necessity of limiting the subjectivity of the experimentalists who analyze and scores the images. A way to bypass the problem of subjective image scoring is the use of high-throughput techniques such as high-content imaging and imaging flow cytometry (e.g. ImageStream™) that permit the rapid acquisition of thousands of biological images and subsequent analysis of these images using dedicated bulk software[52]. While these high-throughput imaging techniques are very powerful and are useful under many circumstances, they lack resolution. As a consequence, their utility is limited to the analysis of bulk cellular/subcellular structures. The application of Kartezio to high resolution, super-resolution, and even electron microscopy images can in principle ensure automated non-subjective analysis with no compromise in terms of resolution.

The ability to reconcile automated image analysis with high resolution imaging is particularly important in the field of nanomedicine, wherein nanotechnologies are applied to diagnostics, therapeutics, vaccines etc. Indeed, characterization of drug vectors and bioactive particles requires the implementation of methods that allow researchers to effectively evaluate and optimize parameters such as particle size and composition. In this sense, Kartezio offers a first stepping stone to address this issue by rapidly creating pipelines for analysis of subcellular structures, such as CTL-derived extracellular particles (Fig. 4). Further optimization of Kartezio is required to make it a reliable "pharmacologist assistant" for validation of nanometer/micrometer size therapeutical preparations.

In summary, Kartezio offers a robust, flexible, and groundbreaking approach to automated image processing. It facilitates the generation of specialist models starting from small or very small datasets. Combined with its low hardware requirements, this confers upon Kartezio an exceptional level of versatility and frugality, broadening its potential applications to include contexts in which large datasets or powerful hardware are unavailable. Although in our study we applied Kartezio to biomedical images, its application can be extended to different fields. It is interesting to note that one of the first application of CGP to image analysis comes from a study in which CGP was applied to images acquired by a Mars rover[11]. Taken together, our study and these pioneering observations strongly suggest that CGP will become increasingly important in the development of CV and its application to different aspects of human life.

## Methods
### Mathematical modeling
Kartezio was developed using Python3[53] programming language and built mainly over NumPy[54], OpenCV[55], Scikit-image[56], and Scikit-learn[36]. For the visualization, images and plots were generated using Matplotlib[57], Seaborn[58], OpenCV[55], and Fiji[59]. Mathematical notations and parameters utilized throughout this manuscript are shown in Table 1.

### Cartesian Genetic Programming: Genotype
CGP considers a specific encoding of the syntactic graph within integer-based genotype (Fig. 1a). Any integer value of the Genotype is called a gene and each node $\mathcal{N}_i$ of the graph is identified with the unique integer value $i$, to which the other graph nodes connect. In CGP, the genotype $\mathcal{G}$ (corresponding to the phenotype $\mathcal{P}$) is an $m$-by-$n$ matrix where $m$ is the number of nodes and is equal to the total

**Table 1 | Mathematical Notations and Parameters Used by Kartezio**

| Kartezio Hyperparameters | |
|---|---|
| **Symbol** | **Description** |
| $\eta \in \mathbb{N}_{\geq 1}$ | Number of functional nodes for the CGP model |
| $\lambda \in \mathbb{N}_{\geq 1}$ | Number of children of the population involved in Evolution Strategy $1+\lambda$ |
| $\mu \in \mathbb{R}[0,1]$ | Mutation probability for functional nodes |
| $\nu \in \mathbb{R}[0,1]$ | Mutation probability for output addresses |
| $\mathcal{K} \in \mathbb{N}_{\geq 1}$ | Number of iterations for the evolution loop |
| **Kartezio Induced Parameters** | |
| $\iota \in \mathbb{N}_{\geq 1}$ | Number of input channels for CGP model; induced by the size of $X$ |
| $o \in \mathbb{N}_{\geq 1}$ | Number of intermediate outputs for CGP model; induced by the arity of the Endpoint |
| $\alpha \in \mathbb{N}_{\geq 1}$ | Number of edges feeding a node (arity); induced from the Function Library $\mathcal{L}$ |
| $\rho \in \mathbb{N}_{\geq 0}$ | Number of parameters used by the functions; induced from the Function Library $\mathcal{L}$ |
| $\varphi \in \mathbb{N}_{\geq 1}$ | Size of the Function Library $\mathcal{L}$ |
| **Kartezio Data** | |
| $x_i \in X$ | One single input (2D channel in case of images) |
| $\hat{z}_i \in \hat{Z}$ | One single intermediate heuristic (2D channel in case of images) |
| $\hat{y}_i \in \hat{Y}$ | One single finale output (any type, usually 2D masks or labels) |
| $y_i \in Y$ | One single annotation (any type, usually 2D masks, annotations) |
| $X : \{x_1 \cdots x_i\} \in \mathcal{X}$ | Input vector of $\iota$ channels corresponding to one entry of a Dataset $\mathcal{D}$ |
| $\hat{Z} : \{\hat{z}_1 \cdots \hat{z}_o\}$ | Intermediate output vector of $o$ heuristics feeding the Endpoint $\mathcal{E}$ |
| $\hat{Y} : \{\hat{y}_1 \cdots \hat{y}_n\}$ | Finale $n$ outputs produced by the Endpoint $\mathcal{E}$ ($n \geq 1$; usually $n=1$) |
| $Y : \{y_1 \cdots y_n\} \in \mathcal{Y}$ | Ground truth annotations ($n \geq 1$; usually $n=1$) |
| $\mathcal{X} : \{X_1 \cdots X_n\}, \mathcal{X} \in \mathcal{D}$ | Training or Test set; list of $n$ ($n \geq 1$) entries $X$ |
| $\mathcal{Y} : \{Y_1 \cdots Y_n\}, \mathcal{Y} \in \mathcal{D}$ | Training or Test set annotations; list of $n$($n \geq 1$) entries $Y$ |
| $\mathcal{D} : \{\mathcal{X}, \mathcal{Y}\}$ | Training or Test Dataset composed of inputs $\mathcal{X}$ and correspond annotations $\mathcal{Y}$ |
| **Kartezio Model Components** | |
| $\mathcal{G}$ | Genotype (vector of integers) |
| $\mathcal{P}$ | Phenotype (directed acyclic graph) |
| $\mathcal{N}$ | Node |
| $\mathcal{L} : \{f_1 \cdots f_\varphi\}$ | Function library composed of $\varphi$ ($\varphi \geq 1$) functions $f$ |
| **Kartezio Operators** | |
| $\delta : \mathcal{G} \to \mathcal{P}$ | Decodes to parse one Genotype to the active Phenotype |
| $\psi$ | Executes the Phenotype function sequence |
| $\mathcal{A}$ | Aggregation (3D specific), function that stack predictions from different image layers |
| $\mathcal{E}$ | Endpoint, acting like a Node to produce the final output $\hat{Y}$ |
| $\mathcal{F}$ | Fitness minimization function, generating a score for the Genotype |

number of functional nodes ($\eta$) and output nodes ($o$). $n$ corresponds to the size of one encoded node (i.e. the number of genes in one node) which relies on 3 values based on the function library $\mathcal{L}$. The first gene corresponds to the unique function index. Then, the next $\alpha$ (called *arity*) genes correspond to the connection of the graph and so the inputs of the function. $\alpha$ is determined by the maximum number of inputs of the functions from the function library $\mathcal{L}$. Lastly, the node is ended with $\rho$ integers corresponding to additional parameters used by

the function. Formally, the genotype can be written as 2 sub-matrices as shown in Equation 1:

$$\mathcal{G} = \begin{bmatrix} \mathcal{N}_{\eta \times n} \\ \mathcal{O}_{o \times n} \end{bmatrix} \in \mathcal{M}_{m \times n}(\mathbb{N}) \text{ where } \begin{matrix} m = \eta + o \\ n = 1 + \alpha + \rho \end{matrix} \quad (1)$$

## Cartesian Genetic Programming: Nodes
A functional node $\mathcal{N}_i$ is a 1-D vector composed of 3 significant segments described as shown in Equation 2:

$$\mathcal{N}_i \in \mathcal{M}_{1 \times n}(\mathbb{N}) : [b_i, C_i, P_i] \text{ where } \begin{matrix} b_i \in [1, \varphi] \\ \forall c \in C_i, c < i \end{matrix} \quad (2)$$

where the integer $b_i$ indexes one of mathematical functions from $\mathcal{L}$ (Supplementary Table 1), $C_i$ are addresses of upstream nodes within the graph that will be used as inputs of the node's function and $P_i$ are constant (but optimized) parameters of the function. All functions from the library must be determinist and must return one 2D image which is identical to the input image in dimensions and type (generally unsigned integer encoded on 8bits, u8). Most of the time, one node operation consists of one function call from the OpenCV Python package. All functions must be image-size independent, which implies that the input and output for a given image is equivalent. However, there is no image size constraint for the model in both training and test datasets.

The first $\iota$ nodes, noted as $X:\{x_1 \cdots x_i\}$ are the receiving inputs (e.g. image channels) while the last $o$ nodes represents the intermediate outputs $\hat{Z} : \{\hat{z}_1 \cdots \hat{z}_o\}$.

An example of genotype $\mathcal{G}[\iota = 3, \eta = 4, o = 2, \alpha = 2, \rho = 2]$ with its nodes is illustrated in Fig. 1.

## Cartesian Genetic Programming: Non-evolvable Preprocessing Node
The first stage of a standard image processing pipeline is the preprocessing or/and formatting of the raw data. Depending on the input images (e.g. transmission light images, fluorescent images, IHC tissues images, etc.) and the expected outcomes (e.g. cell identification, event counting, cell distance and interaction, etc.) needed to address a given biological question, the user can impose a preprocessing of the dataset. When loading the dataset, the preprocessing stage is done on-the-fly and produces a formatted input $X$ for each entry of the dataset. For example, the default behavior for a standard bitmap image is $X_{rgb} : [x_1 = r, x_2 = g, x_3 = b]$. For specific needs of certain datasets, any function can be used for preprocessing (e.g. HSV transformation, pyramidal mean shift filtering, etc.) including those requiring all three channels to operate. This can help evolution by providing important insight given by the domain expert.

## Cartesian Genetic Programming: Non-evolvable Endpoint
We also introduce a new component extending CGP to IS using a determined final function called Endpoint. It is the last processing step before evaluation. It is a substantial bottleneck, not subject to optimization, of the syntactic graph. Moreover, the endpoint is the only node allowing the outcome size and type to change. We use Watershed Transform[20] as the main algorithm to produce IS. In cell segmentation, the cell nuclei are usually stained or visible, so it is possible to deduce good markers for a Marker-Controlled Watershed ($o = 2$)[20]. These two endpoints produce a label map that is then assessed. In SS, the default endpoint is a Threshold (*binary* or *to zero*).

## Cartesian Genetic Programming: Genotype Decoding and Graph Execution
To generate an output from given inputs, the CGP genotype $\mathcal{G}$ is first decoded to produce its phenotype $\mathcal{P}$ (Equation 3). $\mathcal{P}$ is then executed

to produce the intermediary results $\hat{Z}_{\mathcal{G},X}$ (Equation 4) which is provided as the input of the endpoint (Equation 5) to calculate the final output $\hat{Y}_{\mathcal{G},X}$.

$$\mathcal{P} = \delta(\mathcal{G}) \tag{3}$$

$$\hat{Z}_{\mathcal{G},X} = \psi(\mathcal{P},X) \tag{4}$$

$$\hat{Y}_{\mathcal{G},X} = \mathcal{E}(\hat{Z}_{\mathcal{G},X}) \tag{5}$$

When used on 3D images, Equation 3 is used only once to obtain a unique syntactic graph which will be used for all z-sections. Equation 4 is executed for each z-section $X_z$ of the image. All $\hat{Z}_{\mathcal{G},X_z}$ are then aggregated through the $\mathcal{A}$ operator (averaged in the example Equation 6) to obtain the intermediary output $\hat{Z}_{\mathcal{G},X}$, which is in turn feeding the endpoint (Equation 5) to produce the final output $\hat{Y}_{\mathcal{G},X}$ shown in Equation 6.

$$\hat{Y}_{\mathcal{G}},X = \mathcal{E}\left(\mathcal{A}\left(\hat{Z}_{\mathcal{G}},X\right)\right)$$
$$= \mathcal{E}\left(\frac{1}{N} \times \sum_{z=1}^{N} \hat{Z}_{\mathcal{G}},X_z\right) \text{(example of Aggregation by average)} \tag{6}$$

### Cartesian Genetic Programming: Evolution
Kartezio uses the $1+\lambda$ Evolution Strategy (**ES**). A first population ($k = 0$) of genotypes $\mathcal{G}_{k=0}^n$ with $n \in [0,\lambda]$ is initialized randomly and independently. These genotypes then follow an evolutionary loop in which they will first be evaluated by a defined fitness function $\mathcal{F}$ and the best genotype $\mathcal{G}_{k=0}^*$ selected based on evaluation score to produce the next parent genotype. $\mathcal{G}_{k=1}^*$ is then mutated to produce $\lambda$ offspring $\mathcal{G}_{k=1}^n$ with $n \in [1,\lambda]$. The evolutionary loop (evaluation, selection, mutation) is repeated a finite number of $\mathcal{K}$ iterations to produce the final genotype $\mathcal{G}_{\mathcal{K}}^*$. The following section details each evolutionary operator, the $1+\lambda$ ES.

### Cartesian Genetic Programming: Evaluation
To evaluate a genotype $\mathcal{G}$, this genotype is first decoded as describe previously. The resulting graph is executed on each image of the training dataset $\mathcal{X}_{\text{train}}$ to obtain a set of predictions $\hat{Y}_{\mathcal{X}}$. Predictions are compared to the ground truth of the dataset $\mathcal{Y}_{\text{train}}$ using a specific fitness function $\mathcal{F}$ which evaluates the prediction errors. This function is domain-specific (e.g. AP50 for IS or IoU for SS) and must be defined for each task Kartezio is used for. The genotype evaluation fitness$_{\mathcal{G}}$ is finally given by the average of prediction error of each dataset entry (Eq. 7).

$$\text{fitness}_{\mathcal{G}} = \frac{1}{N} \times \sum_{i=1}^{N} \mathcal{F}(\mathcal{Y}_{\text{train}i}, \hat{Y}_{\mathcal{X}_i}) \tag{7}$$

### Cartesian Genetic Programming: Selection
The objective is to minimize the prediction error over many iterations. The best genotype is therefore selected among the offspring and the parent by selecting the smallest prediction error. If an offspring obtains a fitness strictly equal to the current parent, they are distinguished by minimizing the execution time of the graph to promote frugality in term of computational effort.

### Cartesian Genetic Programming: Mutation
The parent generates $\lambda$ new offspring by mutation. Mutations consist of randomly altering random values of the genotype. Kartezio utilizes "Accumulate" mutation behavior proposed by Goldman[60], which implies that mutations accumulate until the active graph changes to

avoid useless evaluations. Random indices ($i,j$) are sampled in the whole functional submatrix: $\mathcal{G}_{i,j}$ gets a new independent value $r$, sampled as follows in Eq. 8:

$$\mathcal{G}_{i,j} = \begin{cases} r \in [1,\varphi] & if\, j = 1 \\ r \in [1, i[ & if\, 1 < j < 1 + \alpha \\ r \in [0,255] & if\, 1 + \alpha < j < 1 + \alpha + \rho \end{cases} \tag{8}$$

The number of altered $\mathcal{G}_{i,j}$ genes is calculated to respect the mutation rate parameter: for instance, if $\mathcal{G}$ has 100 genes and $\mu = 0.1$, then 10 genes will be mutated. Lastly, each output address has a probability to be replaced by a new random value $r \in [1, \iota + \eta]$.

### Cartesian Genetic Programming: Benchmarking
We trained the models with a $\eta = 30$ nodes, $\lambda = 5$ offspring over $\mathcal{K} = 20{,}000$ iterations. If the parent reached a score of 0, the evolution process stopped as no prediction error was made on the training dataset. The mutation probability was $\mu = 0.15$ for the functional nodes and $\nu = 0.2$ for the outputs. 35 independent runs were made for each experiment for statistical purposes. They were run on a bi Intel Xeon @6140 ($2 \times 18$ cores, 2.30 GHz − 3.70 GHz) with 192GB of memory.

### Standardized Testing Using Published Cell Image Library
We compared Kartezio to three state-of-the-art IS algorithms: Cellpose, StarDist and Mask R-CNN. To this end, the dataset and the results for these three algorithms were taken from published data[23,28]. The dataset was composed of 100 images of in vitro cultured neurons stained for phalloidin and DAPI (as shown in Supplementary Fig. 2a, reproduced with permission) and imaged using an epifluorescence microscope[23,28]. This 100-image dataset was split into 89 images for training and 11 images for testing[23]; the training/test split was retained in this manuscript in order to permit direct comparison of Kartezio with Cellpose under standardized conditions. No data augmentation was necessary for our approach.

In this study case, Kartezio was composed of $\iota = 2$ inputs, corresponding to the $a$-tubulin and DAPI channels, and $o = 2$ outputs, corresponding to the mask and markers necessary for the Watershed Transform non-evolvable endpoint. Generated syntactic graphs were evaluated using the Average Precision ($AP = \frac{TP}{TP+FP+FN}$) metric calculated with the standard IoU between the predicted mask and the ground truth. A threshold of $t = 0.5$ was applied to decide the correctness (true-positive) of the predicted mask, as described[23].

Kartezio was benchmarked on different training dataset sizes, ranging from 1 image to 89 images. For each size of dataset, images were selected at random from the original 89 training images. Each trained solution was then evaluated using the previously described dataset of 11 test images[23]. 35 independent training runs were performed, each time randomly selecting different training images from the original set of 89 to control for the impact of training split composition. The AP scores reported in Fig. 2 correspond to the mean and standard deviation of these 35 runs.

### Standardized Testing Using Melanoma Cell Nodules
To extend these results, we tested whether using a specific nonevolvable endpoint (namely the MCW Endpoint) improved Kartezio performance beyond that which was achieved in the absence of a nonevolvable endpoint, using Connected Component to obtain evaluable labels. In the first case (Fig. 2c, green dots), the model must optimize 2 heuristics: the mask and the markers aimed at producing the best IS (see Fig. 1). In the second case (Fig. 2c, magenta dots), the model must learn to distinguish the masks (future labels) directly, which is similar to the CGP-IP version for segmentation. In Fig. 2c, we demonstrate statistically (one-way ANOVA with multiple comparisons) the importance of MCW, on a melanoma nucleus dataset (nuclei stained as orange, Fig. 2, Supplementary Fig. 2b). We also compared the use of

different pre-processing steps (RGB, HSV, or HED) to generalize the comparison. For each of the six conditions, we performed 10 independent runs. Finally, we evaluated the predictions of the best Cellpose pre-trained model (model CPx, on blue channel) in order to estimate the generalist model performance on the same test set.

Among the 30 MCW models, we selected a model with less than 5 active nodes to illustrate Fig. 1. This same model was converted into a Python class (Supplementary Fig. 3) and finally its predictions are illustrated on a test image (512×512) shown in Fig. 2d. In this figure, the first panel shows the manual annotations (green), the second panel shows the Kartezio model predictions (magenta) on top of the annotations and similarly for the last panel with the Cellpose model predictions. An important point to note, the Kartezio model can process more than 100 images like this one (512×512) per second on a single CPU, illustrating the frugality of this approach.

Following this, we assembled four Use Cases with different model inputs and outputs (summarized in Supplementary Table 4) to illustrate Kartezio's versatility.

### Use Case 1 Model Development - Tumor Nodules

24 WSIs from the clinical cohort was assembled for training and testing, including 12 from each clinical site (Institut Universitaire du Cancer de Toulouse and Centre Hospitalier Universitaire de Bordeaux). WSIs corresponded to a 9-level pyramidal image from which we extracted a low resolution thumbnail of dimension 461×1024 pixels (hereafter known as the "training level") using OpenSlide library[61]. These training level images served as the input dataset for Kartezio and were subject to expert pathologist annotation to establish the ground truth for this Use Case.

Images were allocated in a randomized fashion into the training (n = 12) and testing (n = 12) sets (summarized in Supplementary Table 3). Images were preprocessed by transforming R-G-B channels into H-S-V color space (shown in Supplementary Fig. 2c). As summarized in Supplementary Table 4, Kartezio's models were trained with $\iota = 3$ inputs and terminated with a Threshold to zeros endpoint ($o = 1$). Evaluation was made using the simple IoU Fitness function.

An ensemble of 100 models was trained to segment tumor nodules. Each prediction from individual models was normalized (minmax scaler). All 100 predictions were merged by averaging the values pixelwise. All in all, the heat map $\mathcal{H}$ of one slide $X$ is obtained by Eq. 9:

$$\mathcal{H} = \frac{1}{N} \times \sum_{i=1}^{N} \frac{\hat{Y}_{\mathcal{G}_i,X} - \min(\hat{Y}_{\mathcal{G}_i,X})}{\max(\hat{Y}_{\mathcal{G}_i,X}) - \min(\hat{Y}_{\mathcal{G}_i,X})} \text{ where } N = 100 \tag{9}$$

To determine the optimal threshold for the heatmap, we performed incremental evaluation by increasing the final threshold $t$ from 0.0 to 1.0 at a step of 0.01. This provided an evaluation curve (shown in Supplementary Fig. 4) where the optimal threshold was found to be $t = 0.35$. The heatmaps were then thresholded ($t = 0.35$) to produce the heatmaps presented in Fig. 3.

To determine whether the models generated on the low resolution "training level" thumbnails could be upscaled to high resolution images, two strategies were employed using an illustrative example region from the IHC dataset (Fig. 3a, example region indicated with purple box). In a first approach, predictions were performed on the original "training level" image (461×1024 pixels) and the resulting heatmap upsampled using a bilinear resizing function to fit Level 7 (1405×3123 pixels) and Level 5 (5621×12494 pixels) of the image pyramid (Fig. 3c). In a second approach, the ensemble of 100 models trained on the original training level (461×1024) image was applied to Level 7 and Level 5 images to generate new masks without any retrain phase (Fig. 3d).

### Use Case 2 Model Development - Extracellular Particles

Because of the intensity difference between channels (shown in Supplementary Fig. 2d), the dataset was split into two datasets, one to generate a WGA specialist model and one to generate a DiO specialist model. The same procedure was applied to both WGA and DiO datasets. Each dataset included one training image and one testing image of CTL-derived extracellular particles. Training and testing sets were augmented by splitting each image into four quarters. The final post-segmentation analysis was made using the test and training images combined with nine previously unseen images. Use Case 2 dataset preparation is summarized in Supplementary Table 3.

As shown in Supplementary Table 4, Kartezio used $\iota = 1$ channel either for WGA or DiO and $o = 1$ that produces mask. This mask was passed to the Local-Max Watershed non-evolvable endpoint, which consisted of creating a Distance-Transform (DT) map from the mask to find local-max (peaks). Together the peaks and the smoothed gradients of the DT were provided as input to Watershed Transform. Generated syntactic graphs were evaluated as described in the first study case.

Once trained, both models generated masks containing prediction of WGA particle and DiO particle instances. The two masks were merged using the same procedure to evaluate mask prediction to expected ground truth (IoU with a threshold of 0.05).

### Use Case 3 Model Development - CTL Lytic Arsenal

From the original set of 19 images, 5 images were selected for the training dataset and 4 for the test dataset. Each 3D image was composed of 4 channels. Use Case 3 dataset preparation is summarized in Supplementary Table 3. To reduce the computational cost, we resized the images (halved the size). We chose a threshold $t = 0.7$ for the fitness function (AP70) to be more stringent with the predicted mask. To process a 3D image, we provided the z-sections of $\iota = 1$ channel (CD45) to the model, which generates $o = 1$ mask (as summarized in Supplementary Table 4). All the masks were averaged to produce one final mask of the whole 3D image. We then applied the Local-Max Watershed Endpoint to produce the final 2D IS. Cells were segmented from the 19 original images using the best model out of 35 runs. 161 cells were filtered from predicted cells using their pixel area (2000px <area <14000px). For each cell, we calculated intensity features (see below) based on the perforin, granzyme B and CD107a channels (shown in Supplementary Fig. 2e). All the cell's feature vectors were then used to generate the UMAPs[34] of Fig. 4. To this end, we used the following parameters to the UMAP function: dimensionality reduction down to 2 components, using following parameters: n_neighbors=15, min_dist=0.01.

### Use Case 4 Model Development - Lytic Synapses

Four 3D images (using the z-stack function) were acquired. These large images were cut into a 3×3 grid, with a depth interval set for each tile (adapted to the focus). Images with high background noise and culture well edges were removed, thus giving a total of 32 selected images (representative image shown in Supplementary Fig. 2f). We randomly extracted 8 images for the training set and 4 for the test set (summarized in Supplementary Table 3). As shown in Supplementary Table 4, the models handled $\iota = 2$ channels (a-tubulin, DAPI) and used the MCW endpoint (o = 2). Similar to the previous application, each z-section produced one markers map and one mask map, which were averaged individually before being passed to the Endpoint. In that case, the models segmented cells with no distinction of types. Over the 32 images, predicted cells were filtered by pixel area (350 <area <6000). Segmented cells were then classified as either CTLs or target cells by using their intensity feature vector (see below) combined with their area. Unsupervised learning was used for classification using first Principal Component Analysis (n_components = "mle") followed by a Gaussian Mixture Model (GMM) (n_components=2, covariance_type =

"tied"). Finally, CTL/target cell synapses were detected when the Euclidian distance between the centroids of a CTL and a target cell was less than 70px (~ nm). The target cell masks of potential synapses were then dilated (morphological dilation) to assess potential overlapping with CTL mask. A valid CTL/target cell synapse was detected if at least one pixel was overlapping.

## Intensity Feature Vector

This vector contained individual and combined counts of positive pixels, the sum, and the average pixel fluorescence intensities. Combined counts were calculated by first applying a binary threshold to determine the positive pixels within each channel. Each pixel of a given cell mask incremented a counter corresponding to its combination of positive channels. All the counters, augmented with the total count, the sum and the average fluorescence intensity for each channel were gathered into the intensity feature vector. The length $l$ of the vector was determined by the number of channels $C$ shown in Eq. 10:

$$\ell = 2^{\mathcal{C}} + 3 \times \mathcal{C} \tag{10}$$

## Human Samples: Use Case 1

The established melanoma clinical cohort[29] was comprised of patients treated for advanced melanoma both at the Oncodermatology Department of the Institut Universitaire du Cancer de Toulouse (IUCT) and at the Centre Hospitalier Universitaire (CHU) de Bordeaux, who provided written informed consent for all data used. Samples in Toulouse were stored at the CRB Cancer des Hôpitaux de Toulouse collection. In accordance with French law, this cancer collection was declared to the Ministry of Higher Education and Research (DC-2020-4074) and a transfer agreement was obtained (AC-2020-4031) after approbation by ethical committees. Samples in Bordeaux were stored at the Cancer Biobank of CHU Bordeaux collection. In accordance with French law, this cancer collection was declared to the Ministry of Higher Education and Research (DC 2014-2164) and a transfer agreement was obtained (AC-2019-3595) after approbation by ethical committees. Patients with available formalin-fixed, paraffin-embedded (FFPE) tumor blocks at the Department of Pathology were included in the analysis.

## Human Samples: Use Case 2

De-identified peripheral blood samples from healthy adult donors were provided by the Oxford Blood Centre (Oxford Radcliffe Biobank, Research Tissue Bank, REC 19/SC/0173), after obtaining informed consent from each donor, under the Kennedy Institute of Rheumatology ethics agreement number 11-H0711-7, approved by the National Health Service Research Ethics Committee (UK).

## Human Samples: Use Cases 3 and 4

Blood samples were collected and processed following standard ethical procedures after obtaining written informed consent from each donor and approval by the French Ministry of the Research (transfer agreement AC-2020-3971). Approbation by the ethical department of the French Ministry of the Research for the preparation and conservation of cell lines and clones starting from healthy donor human blood samples has been obtained (authorization no. DC-2021-4673).

## Preparation of Extracellular Particles and Single Particle Imaging

For preparation of extracellular particles (EPs), CD8+ human primary T cells were isolated from healthy donor blood with negative selection kits (RosetteSep Human CD8+ T cell Enrichment Cocktail, STEMCELL Technologies, UK) following the manufacturers instruction. CD8+ cells from two different donors were expanded after isolation by addition of anti-CD3/anti-CD28 T-cell activation beads (Dynabeads ThermoFisher

Scientific, UK) at $25\,\mu l/\,10^6$ cells and with 5000 Units/ml recombinant IL-2 for 3 days following the manufacturer's instructions. Subsequently, the magnetic beads were removed and cells were rested for 4 days at $10^6$ cells/ml. For all incubation steps, cells were cultured in RPMI (Gibco) medium supplemented with 10% fetal bovine serum, 1% penicillin/streptomycin, 1% L-glutamine, 1% non-essential amino acids and 50 mM HEPES in cell culture treated flasks at 37 °C, 5% CO2 and 100% humidity. After resting, cells were transferred to fully-supplemented culture medium containing exosome-depleted serum (Thermo Fischer Scientific, UK) and cultured for 48 hours. For EP isolation, the culture medium was subsequently collected (total of 50 ml) and cells were removed by centrifugation for 10 min at 300 g. Larger particles and cell debris were removed by centrifugation for 20 min at 900 g. Subsequently, EPs were collected by centrifugation for 60 min at 100,000 g and at 4 °C. The EP pellet was resuspended in 1 ml phosphate buffered saline (PBS) and stored until further use at −20 °C. Repeated thawing and freezing cycles were avoided.

The EP solution (total protein concentration of 137 µg/ml) was diluted 1:10 with PBS and stained with a final concentration of 40 µg/ml AlexaFluor647-coupled WGA (ThermoFisher Scientific, UK) and 20 µM DiO (ThermoFisher Scientific, UK) for 1 hour at 4 °C in the dark. 150 µl of the EP sample were then deposited onto clean glass coverslip (SCHOTT UK Ldt, Stafford, UK) fixed to flow chambers (sticky- Slide VI 0.4, Ibidi, Thistle Scientific LTD, Glasgow, UK) that were coated with 0.1 mg/ml poly-L-lysine (Sigma Aldrich, UK) for 15 min and washed twice with PBS before addition of EPs. The EPs were incubated for 30 min at 4 °C in the dark to allow for EP binding on the surface. Excess dye and non-bound EPs were washed from the flow channels by washing twice with PBS. Samples were subsequently imaged by TIRFM using an Olympus IX83 inverted microscope (Keymed, Southend-on-Sea, UK) equipped with 405-nm, 488-nm, 561-nm and 640-nm laser lines and a Photometrics Evolve delta EMCCD camera. Images were acquired with a 150×1.45 NA oil-immersion objective.

## Confocal Microscopy: Staining for CTL Lytic Components

Total CD8+ T cells were purified from healthy donor blood samples using the RosetteSep Human CD8+ T Cell Enrichment Cocktail (StemCell Technologies) following the manufacturers instruction. CD8+ T cells were cultured in RPMI 1640 medium supplemented with 5% human AB serum (Institut de Biotechnologies Jacques Boy), 50 µM 2-mercaptoethanol, 10 mM Hepes, 1% MEM-Non-Essential Amino Acids (MEM-NEAA) Solution (Gibco), 1% sodium pyruvate (Sigma-Aldrich), ciprofloxacin (10 µg/ml) (AppliChem), human recombinant interleukin-2 (rIL-2; 100 IU/ml), and human rIL-15 (50 ng/ml) (Miltenyi Biotec). CD8+ T cells were collected 7 days after activation using anti-CD3/anti-CD28 T-cell activation beads (Dynabeads ThermoFisher Scientific) at $2.5\,\mu l/\,10^6$ cells. Cells were washed in serum-free RPMI 1640 and allowed to adhere to poly-L-lysine (Sigma Aldrich) coated slides (Erie Scientific, ER-208B-CE24) at 37 °C, 5% CO$_2$ for 10 min. Cells were then fixed with 3% paraformaldehyde (MP Biomedicals) for 10 min at room temperature, washed twice with PBS and subsequently permeabilized with 0.1% saponin (in PBS/3%BSA/HEPES). Cells were stained in a two-step process with primary antibodies followed by isotype matching secondary antibodies all at 10 µg/ml. Both steps were performed for 1 h at room temperature. The following antibodies were utilized: anti-human perforin mAb (clone δG9, IgG2b, BD 556434; 10 µg/ml) followed by goat anti-mouse IgG2b AlexaFluor555 (Thermofisher A21147; 10 µg/ml), anti-human CD107a rabbit Ab (polyclonal, Abcam ab24170; 5 µg/ml) followed by goat anti-rabbit AlexaFluor647 (Thermofisher A21245; 10 µg/ml), anti-human granzyme B mAb (clone GB11, IgG1, Thermofisher MA1-80734; 10 µg/ml) followed by goat anti-mouse IgG1 AlexaFluor488 (Thermofisher A21121; 10 µg/ml), and anti-human CD45 rat Ab (clone YAML501.4, Thermofisher MA5-17687; 10 µg/ml) followed by goat anti-rat AlexaFluor405 (Thermofisher A48261; 10 µg/ml). The samples were mounted in 90% glycerol-PBS

containing 2.5% DABCO (Sigma). Randomly selected cells were imaged with a LSM 780 confocal microscope equipped with a 63x-NA 1.4 oil immersion Plan-Apochromat objective (Zeiss), zoom 1.0. Images were acquired as z-stacks with an interval of 0.75 μm.

### Confocal Microscopy: Staining for Immunological Synapse Detection

Target cells were either unpulsed or pulsed with 10 μM antigenic peptide for 2 h at 37 °C in RPMI 5% FCS/HEPES and washed three times. Conjugates were formed by 1 min centrifugation at 455 g, incubated for 15 min, gently disrupted and seeded on Poly-L-Lysin coated slides and fixed with 3% paraformaldehyde at 37 °C. Cells were then permeabilized with 0.1% saponin (in PBS/3%BSA/HEPES), and stained with anti-human perforin mAb (10 μg/ml, clone δG9; BD Pharmingen BD 556434) followed by AlexaFluor555 goat anti-mouse IgG2b (10 μg/ml; Thermofisher A21147) and anti-α-tubulin mAb (1 μg/ml, clone DMA1; Sigma Aldrich #T6199) followed by AlexaFluor488 goat anti-mouse IgG1 (10 μg/ml; Thermofisher A21121). The samples were mounted in 90% glycerol-PBS containing 2.5% DABCO (Sigma) supplemented with DAPI (1 μg/ml; Invitrogen) and examined using a LSM 780 (Zeiss) confocal microscope over a 63x Plan-Apochromat objective (1.4 oil). 3D images (using the tile scan and z-stack functions) were acquired.

### Immunohistochemistry

In accordance with previous publications[29], CD8, CD107a and Sox10 were visualized by multiplex IHC for the entire tumor nodule and surrounding tissue. The Discovery ULTRA (Ventana Medical Systems, Innovation Park Drive Tucson, Arizona 85755 USA, ROCHE) was used for automated staining. After dewaxing, tissue slides were heat pre-treated using CC1 buffer (05424569001, ROCHE). Slides were stained using the RUO Discovery Universal procedure (v0.00.0370) in a 3 step protocol with sequential denaturation (CC2 buffer (pH6), at 100 °C, 05279798001, ROCHE). Tissue slides were incubated using primary antibodies in Envision Flex diluent (K800621-2, Agilent Technologies): CD107a [clone D2D11, #9091, Cell Signaling Technology, Inc (diluted 1/30 in Envision Flex Diluent; Agilent Technologies)]; CD8 [clone C8/144B, #M7103, Agilent Technologies (diluted 1/200 in Envision Flex Diluent; Agilent Technologies)]; and Sox10 [clone SP267, #07560389001, ROCHE (provided ready to use)]. Targets were then linked using the OmniMap anti-rabbit (05269679001, ROCHE) and OmniMap anti-mouse (05269652001, ROCHE) HRP conjugated secondary antibodies (all provided ready-to-use). Visualization of the different targets was performed using Discovery Silver (07053649001, ROCHE), Purple (07053983001, ROCHE) and Yellow (07698445001, ROCHE) detection kits. Tissue slides were counterstained using Haematoxylin (05277965001, ROCHE) enhanced by Bluing reagent (05266769001, ROCHE) and mounted with xylene-based mounting medium (Sakura TissuTek Prisma, Sakura Finetek Europe B.V., The Netherlands). IHC slides were then digitized with a Panoramic 250 Flash II digital microscope (3DHISTECH, Budapest, Hungary) equipped with a Zeiss Plan-Apochromat 20x NA 0.8 objective and a CIS VCC-FC60FR19CL 4 megapixels CMOS sensor (unit cell size 5.5 ×5.5) mounted on a 1.6x optical adaptor, to achieve a scan resolution of 0.24 μm/pixel in the final whole slide image. A set of twenty-four whole-slide images from the clinical cohort was assembled for training and testing, including twelve images from each clinical site (Institut Universitaire du Cancer de Toulouse and Centre Hospitalier Universitaire de Bordeaux). Images were allocated in a randomized fashion into the training (n = 12) and testing (n = 12) cohorts prior to analysis.

### Statistics and Reproducibility

In this manuscript, individual models were evolved in parallel using Kartezio and this process produced populations of independently-derived models, each of which is thus considered an independent experiment. The number of models (n) generated in each Use Case is indicated in the corresponding Figure Legend. Statistics reported in this manuscript represent the mean and standard deviation of the population of independent models (of size n) on a given dataset of images, unless otherwise indicated. All statistical analyses were performed using GraphPad Prism 9 and are two-sided unless otherwise indicated. Means were compared using a one-way ANOVA with multiple comparisons. Means compared against a single fixed value were evaluated using a one-sample $t$-test. Biological replicates are indicated where applicable in the corresponding Methods section.

### Software Utilized

Image handling was performed using ImageJ (1.53o), Zen Black (14.0.27.201) and CellSens (XV 3.26). Data was visualized using Matplotlib, Seaborn, and OpenCV (Python packages) as well as GraphPad Prism (9.5.0). Statistical analysis was performed using GraphPad Prism (9.5.0).

### Reporting summary

Further information on research design is available in the Nature Portfolio Reporting Summary linked to this article.

## Data availability

Source data associated with this study including all relevant raw images, datasets and genotypes are provided online with this manuscript as Source Data Files.

## Code availability

Source code is available for non-commercial use on GitHub (https://github.com/KevinCortacero/Kartezio). Kartezio-related scripts are also available on GitHub (https://github.com/KevinCortacero/KartezioPaper).

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

## Acknowledgements

This research has received funding from the European Research Council
(ERC) under the European Union's Horizon 2020 Research and Innova-
tion Programme (Grant agreement No. Syn- 951329; SV and MD). This
work was performed using HPC resources of the CALMIP super-
computing center under the allocation 2022-[P16043]. This work was
supported by the AI Interdisciplinary Institute ANITI, funded by the
French program "Investing for the Future – PIA3" (grant agreement no.
ANR-19-PI3A-0004, SCB). This work was also supported by grants from
the Laboratoire d'Excellence Toulouse Cancer (TOUCAN) (contract
ANR11-LABEX; SV). This work also received support from an ERC Marie
Sklodowska Curie Actions/UK Engineering and Physical Science
Research Council fellowship (E. Pantano/X023907/1, MD and OS). KC
was supported by CARe – Graduate School N°ANR-18-EURE-0003,
managed by the Agence Nationale de la Recherche under the Pro-
gramme Investissements d'Avenir. The funders had no role in prepara-
tion of the manuscript.

## Author contributions

KC designed the research, wrote the code, performed experiments,
analyzed the results, and wrote the paper. BM designed research, ana-
lyzed the results, and wrote the paper. SM provided cellular images and
edited the paper. RK provided cellular images. FL provided cellular
images. GC provided cellular images. NVA provided tissue images. FXF
provided tissue images. LL provided tissue images and pathology con-
sultations. NM provided tissue images. BV provided tissue images. DW
advised on project design. HL advised on project design. OS provided
particle images and edited the paper. MD provided particle images and
edited the paper. SV designed the research, analyzed results, wrote the
paper, and supervised biological/tissue image acquisition. SCB
designed the research, analyzed results, wrote the paper, and super-
vised computational experiments and data analysis.

## Competing interests

The following patent application has been filed by KC, SV, SCB, DW, HL,
and BM.: "**K. Cortacero et al. filed patent; EP 22307041.8**". The other
authors declare no competing interests.

## Additional information

**Supplementary information** The online version contains
supplementary material available at

Salvatore Valitutti or Sylvain Cussat-Blanc.

**Peer review information** *Nature Communications* thanks Una-May
O'Reilly and the other, anonymous, reviewer(s) for their contribution to
the peer review of this work. A peer review file is available.

**Publisher's note** Springer Nature remains neutral with regard to
jurisdictional claims in published maps and institutional affiliations.

[1]Institut National de la Santé et de la Recherche Médicale (INSERM) UMR1037, Centre de Recherche en Cancérologie de Toulouse (CRCT), Toulouse, France.
[2]Centre National de la Recherche Scientifique (CNRS) UMR5071, Toulouse, France. [3]University of Toulouse III - Paul Sabatier, Toulouse, France. [4]Department
of Pathology, Institut Universitaire du Cancer-Oncopole de Toulouse (IUCT), Toulouse, France. [5]Department of Dermatology, IUCT, Toulouse, France. [6]Service
de Pathologie, Centre Hospitalier Universitaire de Bordeaux, Bordeaux, France. [7]INSERM UMR1053 -UMR BaRITOn, Université de Bordeaux, Bordeaux, France.
[8]University of Toulouse - Institut de Recherche en Informatique de Toulouse (IRIT) - UMR5505, Artificial and Natural Intelligence Toulouse Institute,
Toulouse, France. [9]Kennedy Institute of Rheumatology (KIR), Nuffield Department of Orthopedics, Rheumatology and Musculoskeletal Sciences, University of
Oxford, Oxford, UK. [10]Present address: Leibniz Institute for New Materials, 66123 Saarbrücken, Germany. [11]These authors contributed equally: Salvatore
Valitutti, Sylvain Cussat-Blanc. ✉e-mail: salvatore.valitutti@inserm.fr; sylvain.cussat-blanc@irit.fr

