## [Peer Review File · Nature Communications]

Evolutionary design of explainable algorithms for biomedical image segmentationREVIEWER COMMENTS

Reviewer #1 (Remarks to the Author):

The authors present a new machine learning method, which is an extension of Cartesian Genetic Programming (CGP), that can be used to design Computer Vision (CV) image processing pipelines that segment low resolution images. The method, called Kartezio, proceeds with accuracy better or comparable to state of art methods that use much, much more data and do not offer any degree of explainability. It generates short, modular, and easily interpretable pipelines, with small datasets. Models of millions of parameters, that are completely opaque and which have to be trained on large labeled datasets are matched and improved over with a computationally evolved, short, simply-connected pipeline of comprehensible CV functions. It seems infeasible to manually compose such a variety of pipelines of this accuracy so quickly. It would require CV expertise, extensive domain expertise, and hand optimizations.

Kartezio is compared to 3 different state of art Deep Learning (DL) approaches: Cellpose, Mask R-CNN, and StarDist. The comparison uses the dataset that was used in training and evaluating Cellpose and a set of images the authors generated and previously introduced. Kartezio frequently performs better than two (despite training with less data) and is comparable to a third (Cellpose) in accuracy. However, Cellpose is pretrained on 70000 labeled images and fine-tuned on 89. Kartezio is trained with set sizes from 1 to 89. As would be expected, its performance improves as that size increases). This is a noteworthy example of few-shot learning and contrasting model complexity.

The demonstrations, through 4 use cases in microscopy cover both kinds of segmentation -- instance and semantic. The images differ across dimensionality (2D and 3D), imaging type, staining, and scale. They all share the problem of low numbers of labeled images being available. The test accuracies are impressive, given such small training sets which often induce over-specialization and a lack of generalization power.

These results imply that Kartezio is methodologically novel and important.

The authors extend Cartesian Genetic Programming in two ways that others in the field will likely adopt: they introduce non-evolvable nodes that scaffold solutions and allow human expertise to be integrated into a modeling process. Second, they set up so-called "endpoints" which are final transforms, that reduce the search space for syntactic graphs. This latter innovation is transferred from some current DNN-methods.

The authors appear to be an inter-disciplinary team with clinicians, pathologists, and Computer Scientists. This would imply the tasks posed as use-cases will be served well by the method. The method is applicable to other low resolution, CV segmentation tasks as well.

The key distinction of this work is its integration. I am not aware of literature that integrates this approach to modeling (extended CGP), this sort of model improvement (few shot learning and interpretability) with a similar broad set of examples from existing and new clinical datasets.

The appropriate prior CGP work and particularly work on images is cited. One possible additional citation is:

P.C.D. Paris, E.C. Pedrino, M.C. Nicoletti, Automatic learning of image filters using Cartesian genetic programming. *Integr. Comput. Aided Eng.* 22(2), 135–151 (2015).

The concept of evolved or learned pipelines is also introduced in the T-POT Genetic Programming with Sci-Kit Learn project and most recently HuggingGPT: Solving AI Tasks with ChatGPT and its Friends in HuggingFace <https://arxiv.org/abs/2303.17580>

Overall the conclusions, discussion and limitations appear sound. There are a number of methodological details that could be provided to further support claims:

1. A low number of training samples introduces the risk of sample bias, how sensitive is Kartezio to training split composition?
2. Per 1, how were the 11 test images were drawn among the 100?
3. Per 1, was cross-validation was used or why not? E.g. In Fig 1 b, Is the data in 3 splits: train, test, holdout?
4. When the training set size is so low, why were more experiments were not executed? Why not repeat the experiments for every size up to 89, given the method is so fast?
5. How do you pick the hyperparameters of Kartezio? Is it robust to these choices? Are there recommended settings?

The data collection appears ethical. The data pre-processing steps are well documented. The analysis has two parts. The first part compares Kartezio to 3 existing algorithms and the second demonstrates a variety of different use cases.

The X-axis in the plot of Fig 2 should be marked to indicate the dilation from 15,20, 25, 50, 89.

Minor Corrections.

References: Check 6, 14, 15, 37, and 51 and supply more information.

Use Case 4: remove much of text on image acquisition and processing -- staining, patient cohort, immunohistochemistry to supplemental.

In "Kartezio" description

- Relate arity and parameters to node in and out degree.
- The symbols and descriptions appear incomplete
 - L563 Domain for mutation probabilities, are they $[0,1]$?
 - L564 what is x and \hat{z} ? What is list of many X inputs? Is it $X_{\text{set}} \subset X \mid |X_{\text{set}}| < |X|$. Same for Y
 - L565 What are the domains of the symbols? Does the G mean that genome, individual, chromosome are synonyms? What is the domain of ϕ ?
 - What are domains and co-domains of the functions? Only some are listed
- L611 Cite use of Watershed Transform claims
- L614 The examples of how to extend endpoints seems more like discussion than "pure" methods.
- L623 Why not define a merging function to simplify the description notation for 3D?
- L628 Should there be some formal notation for the assigning of a score? E.g. a scoring function?
- L639 Word choice: Compared instead of "confronted"
- L652 Word choice: Changes // accumulate.
- L657 Clarify. Respective?
- L665 $G_0 = G_B$?

Reviewer #2 (Remarks to the Author):

Summary of this paper:

In this paper, the authors propose Kartezio, a computational strategy based on modular Cartesian Genetic Programming. Kartezio is designed to generate image processing pipelines that are fully transparent and easily interpretable. The strategy achieves this by iteratively assembling and parameterizing computer vision functions. The resulting pipelines produced by Kartezio demonstrate comparable precision to state-of-the-art Deep Learning approaches in instance segmentation tasks. However, they require significantly smaller training datasets. This Few-Shot Learning method employed by Kartezio offers notable advantages in terms of flexibility, speed, and functionality. To evaluate the effectiveness of Kartezio, the authors conducted experiments on a variety of semantic and instance segmentation problems. The utility of Kartezio was demonstrated across diverse images, including multiplexed tissue histopathology images and high-resolution microscopy images. Overall, Kartezio presents a promising approach that combines interpretability, performance, and efficiency in image processing tasks. The experiments highlight its potential in addressing challenging segmentation problems across different image domains.

Strengths:

This work is based on the traditional algorithm CGP (Cartesian Genetic Programming), which may not always yield segmentation results that outperform state-of-the-art deep learning methods. However, it possesses several notable advantages. Firstly, it is less dependent on the size of the dataset, which can be beneficial when working with limited data availability. Secondly, the generated segmentation pipelines are highly interpretable and can be easily understood by human experts. This interpretability aspect opens up possibilities for assisting researchers in automated data construction and evaluation tasks, potentially improving the overall workflow in the field. The experiments conducted in this work are relatively comprehensive, providing a thorough evaluation of the proposed method.

Weaknesses:

Major concerns:

1. The method has several non-evolvable components, including preprocessing nodes and endpoints. The authors claim that these parts are flexible choices made by users based on the task and data. However, this approach seems similar to the practice mentioned in lines 64 to 66 and may introduce subjective biases.

2. Related to the previous point, since preprocessing nodes and endpoints are non-evolvable, how should users select the most suitable ones? What criteria did the authors use in each experiment to make these choices?
3. If the selection of these non-evolvable components is automated, does the method still have an advantage over state-of-the-art deep learning methods in terms of time and computational requirements (Gflops) when achieving similar performance with the same amount of data?
4. How were the 42 functions in the method selected? Is there a possibility of redundancy among them?
5. The method proposes numerous hyperparameters, including the number of nodes (η), the number of children (λ), mutation probability, etc. The authors' ablation study on these hyperparameters is not comprehensive enough. They did not explore the impact of varying these hyperparameters on the method, such as whether it would require more iterations to achieve optimal results or even lead to unreliable results.
6. Related to the previous point, why was the benchmark η not set to the same value as the number of functions in the library, i.e., 42?
7. Line 682 states, "Whole-slide high-resolution images were first saved at their lowest resolution level, strongly reducing image size." Whole-slide images (WSIs) have extremely large sizes, often containing billions of pixels per image. How were they downsampled to the lowest resolution level, and what is the size of the lowest level? Does such a significant downsampling operation lead to inaccurate segmentation boundaries?
8. From a CGP-based perspective, there is no comparable work for biomedical segmentation. However, in comparison to deep learning-based methods, the article lacks a comparison with state-of-the-art (SOTA) methods [1].

[1] Chen, Shengcong, et al. "CPP-net: Context-aware polygon proposal network for nucleus segmentation." *IEEE Transactions on Image Processing* (2023).

9. The segmentation of tumor nodules lacks comparison with recent WSI segmentation method [2].

[2] Chikontwe, Philip, et al. "Weakly supervised segmentation on neural compressed histopathology with self-equivariant regularization." *Medical Image Analysis* 80 (2022): 102482.

Minor issues:

1. The paper contains a few minor errors, including but not limited to:

- (1) Line 77: Missing a comma after "(IS)".
- (2) Line 1037: In Figure 1, the caption "A,b)" should be "(a,b)".

2. The paper provides details of the experiments and promises to release the code. Although a corresponding repository can already be found on GitHub, the code has not been made public yet. It is highly recommended to make the code publicly available as it will greatly assist readers in reproducing the experiments and further advancing the research

Response to Reviewers

Reviewer #1 (Remarks to the Author):

The authors present a new machine learning method, which is an extension of Cartesian Genetic Programming (CGP), that can be used to design Computer Vision (CV) image processing pipelines that segment low resolution images. The method, called Kartezio, proceeds with accuracy better or comparable to state of art methods that use much, much more data and do not offer any degree of explainability. It generates short, modular, and easily interpretable pipelines, with small datasets. Models of millions of parameters, that are completely opaque and which have to be trained on large labeled datasets are matched and improved over with a computationally evolved, short, simply-connected pipeline of comprehensible CV functions. It seems infeasible to manually compose such a variety of pipelines of this accuracy so quickly. It would require CV expertise, extensive domain expertise, and hand optimizations.

We thank the Reviewer for their supportive assessment of our manuscript.

Kartezio is compared to 3 different state of art Deep Learning (DL) approaches: Cellpose, Mask R-CNN, and StarDist. The comparison uses the dataset that was used in training and evaluating Cellpose and a set of images the authors generated and previously introduced. Kartezio frequently performs better than two (despite training with less data) and is comparable to a third (Cellpose) in accuracy. However, Cellpose is pretrained on 70000 labeled images and fine-tuned on 89. Kartezio is trained with set sizes from 1 to 89. As would be expected, its performance improves as that size increases). This is a noteworthy example of few-shot learning and contrasting model complexity. The demonstrations, through 4 use cases in microscopy cover both kinds of segmentation -- instance and semantic. The images differ across dimensionality (2D and 3D), imaging type, staining, and scale. They all share the problem of low numbers of labeled images being available. The test accuracies are impressive, given such small training sets which often induce over-specialization and a lack of generalization power.

These results imply that Kartezio is methodologically novel and important.

We thank the Reviewer for this assessment.

The authors extend Cartesian Genetic Programming in two ways that others in the field will likely adopt: they introduce non-evolvable nodes that scaffold solutions and allow human expertise to be integrated into a modeling process. Second, they set up so-called "endpoints" which are final transforms, that reduce the search space for syntactic graphs. This latter innovation is transferred from some current DNN-methods. The authors appear to be an interdisciplinary team with clinicians, pathologists, and Computer Scientists. This would imply the tasks posed as use-cases will be served well by the method. The method is applicable to other low resolution, CV segmentation tasks as well.

The key distinction of this work is its integration. I am not aware of literature that integrates this approach to modeling (extended CGP), this sort of model improvement (few shot learning and interpretability) with a similar broad set of examples from existing and new clinical datasets.

The appropriate prior CGP work and particularly work on images is cited.

One possible additional citation is:

P.C.D. Paris, E.C. Pedrino, M.C. Nicoletti, Automatic learning of image filters using Cartesian genetic programming. *Integr. Comput. Aided Eng.* 22(2), 135–151 (2015).

We thank the Reviewer for highlighting this important work and have integrated it into **Line 98**.

The concept of evolved or learned pipelines is also introduced in the T-POT Genetic Programming with Sci-Kit Learn project and most recently HuggingGPT: Solving AI Tasks with ChatGPT and its Friends in HuggingFace <https://arxiv.org/abs/2303.17580>

Multiple exciting advances in Large Language Models with relevance to AI tasks have indeed been made contemporaneously with the submission of this manuscript. We have added a paragraph in the **Discussion [Lines 512-523]** to highlight these findings and contextualize their implications for this work.

Overall the conclusions, discussion and limitations appear sound. There are a number of methodological details that could be provided to further support claims:

1. A low number of training samples introduces the risk of sample bias, how sensitive is Kartezio to training split composition?

We agree that with such small datasets, the final evolved pipeline can be sensitive to the train/test split composition. For this reason, as shown in **Figure 2a** and **2b**, we performed 35 independent training runs, each time randomly selecting different training images from the original set of 89 images to control for this impact. The Average Precision (AP) scores reported in **Figure 2** correspond to the mean and standard deviation of these 35 runs. As reported, the low standard deviation obtained reveals the modest impact of train/test split in this application. As shown in **Lines 780-784**, we have refined the text of the Methods section to better explain the evaluation of our method.

2. Per 1, how were the 11 test images were drawn among the 100?

The original Cellpose paper provided the train/test split, which was retained in our manuscript in order to directly compare the performance of Cellpose and Kartezio. For Kartezio's training dataset, we randomly sampled the indicated number of images from Cellpose's training dataset, 35 independent times. Each model was then evaluated on the 11-image test set provided in the original Cellpose manuscript. We have updated the manuscript to more accurately reflect this **[Lines 156-158 and 768-771]**.

3. Per 1, was cross-validation was used or why not? E.g. In **Fig 1b**, Is the data in 3 splits: train, test, hold-out?

We preferred a train/test split instead of cross-validation, as reports in the literature describe data leakage on small datasets (see <https://www.nature.com/articles/s41746-022-00592-y>). We have no hold-out in the paper, but we agree it will be necessary to have such a set in a final application to evaluate the best model among all those trained on a new unseen dataset.

4. When the training set size is so low, why were more experiments were not executed? Why not repeat the experiments for every size up to 89, given the method is so fast?

Each training is indeed not computationally expensive, especially when the dataset contains very few images. In its current iteration, Kartezio is designed to run multiple evolutions in parallel (with images processed sequentially) rather than a single evolution processing many images in parallel. For this reason, instead of evolving Kartezio for all possible dataset sizes, we preferred to multiply independent runs, with different sampling (35 for each dataset size tested) to evaluate the stability of our approach. To refine **Figure 2a** in line with the Reviewer's suggestion, we added 5 additional dataset sizes (consisting of 30, 40, 60, 70 and 80 images) and rescaled the X-axis. We have also added the additional data into **Extended Data Table 2**. We believe these new points are sufficient to define the behavior of Kartezio on larger dataset sizes, but of course we are willing to compute additional datapoints for the manuscript if the Reviewers feel it is imperative.

5. How do you pick the hyperparameters of Kartezio? Is it robust to these choices? Are there recommended settings?

The hyperparameters currently used in our approach (which were kept consistent for all four Use Cases) represent commonly used hyperparameters in the CGP literature, and these could be practically utilized as default parameters or recommended settings for *non-expert users* of Kartezio. However, we envision that *expert users* may wish to modify the hyperparameters to personalize the model to their needs and we agree that a complete hyperparameter study is necessary to understand their impact on the evolution of CV pipelines. We are currently conducting this study and are planning to submit a paper as soon as it is completed.

For the Reviewer's interest, we have included below our preliminary results for four hyperparameters (**Figure A**). Our initial results show that the number of children λ and the number of nodes η do not substantially impact model fitness within the acceptable ranges of values. However, as is known in the literature, the mutation rates μ and u must remain low to optimize performance. The hyperparameters used in our manuscript (which were kept consistent through the current paper) are indicated with a green square. These preliminary data can be added in supplementary material upon request.

We have included an additional discussion in the manuscript highlighting the selection of hyperparameters and the possibility of their modification by expert users [**Lines 581-587**].

Figure A: Impact of Four Common Hyperparameters on Kartezio (Reviewers only)

The Cellpose dataset was utilized to test the impact of **(a)** the number of children λ , **(b)** the number of nodes, **(c)** the mutation probability for nodes μ and **(d)** the mutation probability for outputs across a range of values, for a total of 756 different hyperparameter combinations. For each set of hyperparameter values, Kartezio was trained on eight images drawn at random from the Cellpose training dataset, with five runs performed per condition. The five pipelines thus generated were tested on the Cellpose 11-image test dataset as previously described. AP50 fitness score (*top*), number of active graph nodes (*middle*) and mean processing time (*bottom*) were quantified for each hyperparameter value. For each value of a given hyperparameter on the x-axis, the associated mean and standard deviation represent aggregate data from all other hyperparameters. Values inscribed in green squares represent the hyperparameter values used in the current manuscript.

The data collection appears ethical. The data pre-processing steps are well documented. The analysis has two parts. The first part compares Kartezio to 3 existing algorithms and the second demonstrates a variety of different use cases.

6. The X-axis in the plot of Fig 2 should be marked to indicate the dilation from 15,20, 25, 50, 89.

Thank you for this remark. In addition to doing so, we have added the extra data points as requested above.

Minor Corrections

Thank you for these corrections. The references have been updated as requested, and the methods section improved to include the requested details. Minor changes have been addressed in the text in orange. Additional clarifying details have been provided below.

- References: Check 6, 14, 15, 37, and 51 and supply more information **Done**
- Use Case 4: remove much of text on image acquisition and processing -- staining, patient cohort, immunohistochemistry to supplemental **This text related to multiple Use Cases, so we have subdivided the Methods section more appropriately.**
- In "Kartezio" description **This has been corrected.**
 - Relate arity and parameters to node in and out degree.
 - The symbols and descriptions appear incomplete
- L563 Domain for mutation probabilities, are they [0,1]? **Yes, this has been updated.**
- L564 what is x and \hat{z} ? What is list of many X inputs? Is it $X_{\text{set}} \subset X$ $|X_{\text{set}}| < |X|$. Same for Y **This has been clarified.**
- L565 What are the domains of the symbols? Does the G mean that genome, individual, chromosome are synonyms? What is the domain of ϕ ? **We have harmonized the nomenclature in the text to use only genotype for consistency.**
- What are domains and co-domains of the functions? Only some are listed. **The Extended Data Table 1 has been updated to include all domains and co-domains of functions.**
- L611 Cite use of Watershed Transform claims **Done**
- L614 The examples of how to extend endpoints seems more like discussion than "pure" methods. **This has been removed.**
- L623 Why not define a merging function to simplify the description notation for 3D? **We have introduced as an aggregation operation.**
- L628 Should there be some formal notation for the assigning of a score? E.g. a scoring function? **We have introduced this function as suggested.**
- L639 Word choice: Compared instead of "confronted" **Done**
- L652 Word choice: Changes // accumulate. **Done**
- L657 Clarify. Respective? **Done**
- L665 Go = GB? **Done**

Reviewer #2 (Remarks to the Author):

Summary of this paper:

In this paper, the authors propose Kartezio, a computational strategy based on modular Cartesian Genetic Programming. Kartezio is designed to generate image processing pipelines that are fully transparent and easily interpretable. The strategy achieves this by iteratively assembling and parameterizing computer vision functions. The resulting pipelines produced by Kartezio demonstrate comparable precision to state-of-the-art Deep Learning approaches in instance segmentation tasks. However, they require significantly smaller training datasets. This Few-Shot Learning method employed by Kartezio offers notable advantages in terms of flexibility, speed, and functionality. To evaluate the effectiveness of Kartezio, the authors conducted experiments on a variety of semantic and instance segmentation problems. The utility of Kartezio was demonstrated across diverse images, including multiplexed tissue histopathology images and high-resolution microscopy images. Overall, Kartezio presents a promising approach that combines interpretability, performance, and efficiency in image processing tasks. The experiments highlight its potential in addressing challenging segmentation problems across different image domains.

We thank the Reviewer for their highly supportive assessment of our manuscript.

Strengths:

This work is based on the traditional algorithm CGP (Cartesian Genetic Programming), which may not always yield segmentation results that outperform state-of-the-art deep learning methods. However, it possesses several notable advantages. Firstly, it is less dependent on the size of the dataset, which can be beneficial when working with limited data availability. Secondly, the generated segmentation pipelines are highly interpretable and can be easily understood by human experts. This interpretability aspect opens up possibilities for assisting researchers in automated data construction and evaluation tasks, potentially improving the overall workflow in the field. The experiments conducted in this work are relatively comprehensive, providing a thorough evaluation of the proposed method.

We thank the Reviewer for their insightful comments.

Weaknesses:

1. The method has several non-evolvable components, including preprocessing nodes and endpoints. The authors claim that these parts are flexible choices made by users based on the task and data. However, this approach seems similar to the practice mentioned in lines 64 to 66 and may introduce subjective biases.

We absolutely agree with the Reviewer that the expert selection of non-evolvable components by nature introduces a subjective element into the image analysis process. The impact of these individual subjective decisions on the performance of the model can be objectively assessed (as shown in Fig. 2), which enables the validation of this approach and provides a mechanism to minimize subjective biases. This approach is consistent with the many contemporary Deep Learning approaches that integrate pre- and post-processing functions to improve their performance, and herein we propose an equivalent solution in evolutionary computation. We believe that adding non-evolvable nodes is an important

mechanism for integrating human knowledge, and to support this interpretation, we provide evidence (shown in **Fig. 2c**) that pipelines given an appropriate non-evolvable endpoint significantly outperform those with no endpoint indicated. We have discussed this in **Lines 600-603**.

2. Related to the previous point, since preprocessing nodes and endpoints are non-evolvable, how should users select the most suitable ones? What criteria did the authors use in each experiment to make these choices?

This is an excellent question. We suggest that non-evolvable nodes should initially be selected by *expert users* based upon the input images and matched to the specific objective of the image processing task as described in the **Methods** section (“Non-evolvable Preprocessing Node” and “Non-evolvable Endpoint” subsections). In this manuscript, the selection of preprocessing nodes was primarily guided by the nature of the input image, whereas the selection of non-evolvable endpoints was guided by the task objective (e.g. Marker Controlled Watershed for IS). Importantly, these expert-chosen non-evolvable nodes can be objectively validated through side-by-side comparison to other options (similar to the analysis we performed in **Fig. 2c** comparing HSV, HED, and RGB transformation) or in comparison to the absence of a defined node (shown in **Fig. 2c**, comparing Marker Controlled Watershed to No Endpoint).

Finally, validated non-evolvable nodes can serve as suggested default settings for *non-expert users*, who can choose from a list of recommended non-evolvable nodes based on their input image and task objective. We have added a detailed paragraph outlining the selection of pre- and post-processing nodes in the **Discussion [Lines 588-599]**.

3. If the selection of these non-evolvable components is automated, does the method still have an advantage over state-of-the-art deep learning methods in terms of time and computational requirements (Gflops) when achieving similar performance with the same amount of data?

In the current form of Kartezio, the selection of non-evolvable components is not automated, as it is performed by experts in image analysis as described above. However, automation of the selection of non-evolvable components as the Reviewer suggests is a powerful approach and the object of ongoing investigations by our research group. Evaluation in Gflops is currently difficult to estimate since Kartezio is designed to run on CPU while Deep Learning uses GPUs. However, we believe this is a very interesting comparison to make in order to evaluate the frugality of our approach (with and without non-evolvable nodes or with automated selection of non-evolvable components) compared to Deep Learning. We are planning to conduct such a study in the near future.

4. How were the 42 functions in the method selected? Is there a possibility of redundancy among them?

We agree that the function library is fundamental to achieving good results with Genetic Programming, as neural architecture is in Deep Learning. Developer experience in Genetic Programming and evolutionary computation is currently necessary to identify the best possible function set and find an appropriate balance between the library size (related to the search space size) and the coverage of these functions for a given application.

For the image processing applications in this manuscript, the function library represents a comprehensive expert-curated list of the most common image processing functions in the literature, and as such certain redundancies between similar common functions are not unexpected. The existing list was designed to maximize the diversity of functions available to

Kartezio while ensuring that all key types of function were represented, including: morphological (e.g. Erosion, Dilation), arithmetic (e.g. Add, Sub, Square), logical (e.g. And, Or, Not), and edge detection (e.g. Laplacian, Sobel) among others. We have amended the **Discussion** to include these important considerations [**Lines 559-575**].

As shown in our response to **Comment #8** below, new functions can also be added to the function library by expert users to fit specific tasks and this can improve the performance of the model (**Figure B**).

Ongoing syntactic graph studies are currently underway within our research group to determine the frequency with which individual functions are represented in final pipelines and the propensity of given function pairs (bigrams) or triplets (trigrams) to be used together (e.g. Gaussian Blur followed by Sobel Filter). Clusters of functions typically used together could be entered into the function library as new functions, and rarely used or redundant functions can be removed. These decisions can be made based upon real data derived from the analysis of large numbers of generated pipelines. These data-driven insights will enable the existing image processing function library to be fine-tuned, and expert users have the option to further refine the function library, thus providing an additional opportunity for human knowledge to be integrated with the machine learning task. As mentioned in the **Discussion [Line 577]**, additional research may also make it possible to automatically select/build these functions and/or propose a methodology to build such a library.

5. The method proposes numerous hyperparameters, including the number of nodes (η), the number of children (λ), mutation probability, etc. The authors' ablation study on these hyperparameters is not comprehensive enough. They did not explore the impact of varying these hyperparameters on the method, such as whether it would require more iterations to achieve optimal results or even lead to unreliable results.

We fully agree with both the Reviewers that hyperparameter selection is an important consideration and we are currently conducting a follow-up hyperparameter study to address this. Our initial results show that the values used in the manuscript (taken from CGP literature and experience of our research group; indicated in green boxes in **Figure A** above) are optimal or close to optimal for the four hyperparameters tested. Please refer to our response to **Reviewer #1, Comment 5** and the associated **Figure A** for full details.

6. Related to the previous point, why was the benchmark η not set to the same value as the number of functions in the library, i.e., 42?

In our approach, each function in the library could be used once, more than once, or not at all, and as such it was not imperative to match the number of nodes to the number of functions in a 1-to-1 fashion. The number of nodes η allowed in the pipeline was not necessarily guided by the number of functions in the library but rather to the expected size of the pipeline necessary to analyze the image. In CGP, the genotype is a fixed size in order to avoid graph bloating, which is a very common problem in genetic programming. Keeping a fixed size of genotype ensures that the pipeline remains relatively small and of controllable size. Our data suggests that providing more internal nodes might increase the size of the generated pipelines without a corresponding improvement in term of efficiency. As shown above in **Figure A**, increasing the number of nodes from the current size (30 nodes) up to 100 nodes increases the processing time but fitness scores remain constant.

7. Line 682 states, "Whole-slide high-resolution images were first saved at their lowest resolution level, strongly reducing image size." Whole-slide images (WSIs) have extremely large sizes, often containing billions of pixels per image. How were they downsampled to the lowest resolution level, and what is the size of the lowest level? Does such a significant downsampling operation lead to inaccurate segmentation boundaries?

We have addressed these interesting questions in two parts:

Firstly, we have re-written the corresponding section of the **Methods** to clarify the downsampling processes we used in **Use Case 1 [Lines 814-817]**. In brief, for each melanoma nodule, we began with the WSI (corresponding to a 9-level pyramidal image) and extracted a low resolution thumbnail of size 461x1024 pixels (hereafter known as the "training level") using OpenSlide library. These training level images served as the input dataset for Kartezio, and likewise expert annotations were made on these images.

Secondly, this meaningful Reviewer suggestion has inspired the modification of **Fig. 3** and the creation of an additional supplemental figure (**Extended Data Fig. 4**, described in **Methods, Lines 832-839**) to address the impact of resolution on Kartezio's performance on semantic segmentation tasks.

We deployed two different approaches to test this, using a melanoma nodule from Use Case 1 as an illustrative example. Importantly, neither approach involve new training.

- 1) In the first approach, we performed predictions on the original "training level" image (461x1024 pixels) and upsampled the resulting heatmap using a bilinear resizing function to fit Level 7 (1405x3123 pixels) and Level 5 (5621x12494 pixels) of the image pyramid. This is shown in the new **Fig. 3c**.
- 2) In the second approach, we repurposed the ensemble of 100 models trained on the training level (461x1024) image and applied these models to images of increasingly high resolution (Level 7 and Level 5) to generate new masks. This is shown in the new **Fig. 3d**.

Our new data convincingly illustrate that the downsampling operation does not lead to inaccurate segmentation boundaries at higher resolutions, but rather Kartezio retains the ability to perform well on semantic segmentation tasks on much higher resolution images than those on which it was trained.

The identification of the optimal threshold value was an important consideration in these experiments. As such, in **Extended Data Fig. 4**, we provide a quantitative assessment of fitness as a function of heatmap threshold (described in **Methods, Lines 828-831**). Using the training datasets of melanoma nodules, we determined that the optimal threshold value was $t = 0.35$, and this threshold was utilized throughout the new **Fig. 3**, including the high resolution examples provided in **Fig. 3c-d**.

These important new data show that training on high resolution images (which is computationally expensive and time-consuming) is not necessary for Kartezio to retain its efficacy, and these findings are presented in the **Results** section [**Lines 282-299**].

8. From a CGP-based perspective, there is no comparable work for biomedical segmentation. However, in comparison to deep learning-based methods, the article lacks a comparison with state-of-the-art (SOTA) methods [1].

[1] Chen, Shengcong, *et al.* "CPP-net: Context-aware polygon proposal network for nucleus segmentation." *IEEE Transactions on Image Processing* (2023).

In accordance with the Reviewer's suggestion, we compared the performance of Kartezio with CPP-net and others on a nucleus segmentation task using the BBBC006v1 dataset from Chen *et al*, 2023. The whole set comprises 768 images, and herein we utilized the split provided by the Python script (May 2023) of Chen *et al* with 538 images for training and 115 for testing; in Chen *et al* (2023), the remaining 115 were utilized for validation.

We first undertook this segmentation task using the identical Kartezio setup employed throughout our manuscript (herein referred to as **Setup 1**), training Kartezio on 5, 10, and 20 images. This yielded scores on the test set of AP50 = 0.794 ± 0.037 (5 images), AP50 = 0.811 ± 0.030 (10 images), and AP50 = 0.822 ± 0.022 (20 images). Since the Chen *et al* manuscript evaluated state-of-the-art methods using multiple metrics, we provide results using three different metrics (AP50, AP90 and IoU, known as AJI in Chen *et al*), the results of which are shown in **Figure Bi-iii** below (*for Reviewers only*, **Setup 1** shown in blue).

Since we recognized that **Setup 1** could be further optimized for the current task, we decided to draw upon one of the strengths of Kartezio – i.e. the ability to integrate human knowledge - and updated the setup to more appropriately fit the nucleus segmentation task. We therefore relaunched the experiments using 5, 10, and 20 images with new functions: embossing, normalization, Otsu threshold, pyrup/pyrdown, and Kirsch filter. Furthermore, we created a fitness function combining AP50 and IoU. Collectively, this new setup is referred to as **Setup 2**. As shown in **Figure Bi-iii**, the results from **Setup 2** (shown in orange) offer a consistent improvement in performance compared to the original **Setup 1**, in all three conditions tested (5, 10, and 20 images) for all three metrics (AP50, AP90, and IoU), with a particularly drastic improvement shown in the AP90 metric.

We then compared the performance of Kartezio using **Setup 2** to the state-of-the-art DL methods reported in the Chen *et al* paper. As shown in **Figure Biv-vi**, Kartezio's best-performing pipeline (from only 20 runs) performed comparably to state-of-the-art DL methods and outperformed many of them in terms of AP90 and IoU metrics, even when the size of Kartezio's training dataset was 99% smaller than that of the DL methods (e.g. 5 images in Kartezio's training dataset versus 538 for the DL methods).

This approach highlights how easily Kartezio's parameters can be adapted to new segmentation tasks by expert users, in order to keep pace with state-of-the-art DL models while preserving the benefits of few-shot learning and full interpretability. Nonetheless, we propose **Figure B** for the Reviewers' interest only and suggest that it not be included in the manuscript (which has a single harmonized function library for all Use Cases) in order to avoid complication of the manuscript.

As an additional consideration, it is worth noting that the Ground Truth segmentation of the BBBC006v1 image dataset (<https://bbbc.broadinstitute.org/BBBC006>, Ground Truth section) was performed using an automated image processing pipeline that was prone to certain known limitations (e.g. over-segmentation). An example of this is provided in **Figure Bvii-ix**, wherein an individual large nucleus is segmented into 5 segments by the automated algorithm. Kartezio, however, correctly identifies the single large nucleus; the performance of the different ML algorithms may be affected differently and to various degrees depending upon the extent to which they learn the errors in the ground truth dataset. In the future, Kartezio may also serve as a useful tool to provide Ground Truth annotations on which other models may be trained, including deep neural networks (DNN).

Figure B: Performance of Kartezio-generated models on BBBC006v1 dataset (*Reviewers only*)

(i-iii) Kartezio was trained on 5, 10 or 20 images, derived from the training set of the BBBC006 dataset (Chen *et al*, 2023) either using the identical function library, fitness function, and hyperparameters utilized in our existing manuscript (collectively referred to as **Setup 1**, shown in blue) or using an updated function library and fitness function (**Setup 2**, shown in orange). Using both setups, Kartezio’s performance on a previously unseen test set of 115 images from the BBBC006v1 dataset is shown using (i) AP50, (ii) AP90, or (iii) IoU as the metric for model evaluation. Data shown represent the mean \pm standard deviation of $n = 20$ pipelines for the indicated number of training images. (iv-vi) Performance of the best Kartezio pipeline (from 20 runs) compared against state-of-the-art models. Dotted lines indicate the reported results obtained by each of the other methods (a = CPP-Net; b = StarDist; c = HoVer-Net; d = KeypointGraph; e = InstanceEmbedding; f = Watershed; g = Ellipse Fitting). (vii-ix) Example of over-segmentation of test image by the automated Ground Truth algorithm (viii, white arrow) correctly segmented by Kartezio (ix, white arrow).

9. The segmentation of tumor nodules lacks comparison with recent WSI segmentation method [2].

[2] Chikontwe, Philip, *et al.* "Weakly supervised segmentation on neural compressed histopathology with self-equivariant regularization." *Medical Image Analysis* 80 (2022): 102482.

We agree that several interesting DL approaches to tumor nodule segmentation have recently emerged in the literature. Indeed, WSI segmentation is of interest to many researchers and we are currently evaluating the capacity of our approach in comparison to the recent DL approaches to WSI segmentation.

As described, the weakly supervised WSI segmentation approach detailed by Chikontwe *et al* requires image-level labels generated by expert pathologist annotation (i.e. Level 0 of our 9-level image pyramid, as described above in **Response #7**) to train the segmentation algorithms. However, such image-level annotation is not currently available and would be extremely time- and labour-intensive to generate for our tumor nodules due to the large size and complex nature of the melanoma nodules. Nonetheless, we are currently in the process of developing a human-in-the-loop version of Kartezio in which the intermediate results from Kartezio-generated pipelines are displayed while the expert amends the automatically generated annotation (as described in the **Discussion**). This human-in-the-loop approach will vastly decrease the time and labour required for image-level annotations and we believe this is a crucial step that we must achieve before being able to adequately test Kartezio side by side with weakly supervised WSI approaches. These studies are currently underway on our team with the aim of leveraging Kartezio's proven generalizability, which allows us to minimize both annotations and cost while optimizing performance on tumor nodule segmentation tasks.

Minor issues:

1. The paper contains a few minor errors, including but not limited to:

(1)Line 77: Missing a comma after "(IS)" **Done**

(2)(2) Line 1037: In Figure 1, the caption "A,b)" should be "(a,b)" **Done**

2. The paper provides details of the experiments and promises to release the code. Although a corresponding repository can already be found on GitHub, the code has not been made public yet. It is highly recommended to make the code publicly available as it will greatly assist readers in reproducing the experiments and further advancing the research.

We fully agree and the code will be made available to the public upon acceptance of the manuscript. The source code is currently available to Reviewers (kartezio.zip) along with two demo datasets (workspace.zip) used in the paper and a README file detailing instructions for installation and use.

REVIEWERS' COMMENTS

Reviewer #1 (Remarks to the Author):

The authors have provided satisfactory responses and edits to my first round remarks. For me, the paper is ready to go.

Reviewer #2 (Remarks to the Author):

The author's response and revised manuscript have addressed most of my concerns. However, I still recommend that the authors consider integrating the selection of non-evolvable nodes for different tasks, as described by them, into their pipeline in the future. Additionally, I suggest that they include a comparison with deep learning-based methods (e.g., in response to comment 8) in their GitHub repository, which would provide readers with a clearer understanding of the approach.

REVIEWERS' COMMENTS

Reviewer #1 (Remarks to the Author):

The authors have provided satisfactory responses and edits to my first round remarks. For me, the paper is ready to go.

We thank the Reviewer for her support of this manuscript.

Reviewer #2 (Remarks to the Author):

The author's response and revised manuscript have addressed most of my concerns. However, I still recommend that the authors consider integrating the selection of non-evolvable nodes for different tasks, as described by them, into their pipeline in the future.

We absolutely agree with the Reviewer regarding the integration of the automated selection of non-evolvable nodes into the pipeline in the future. This will be part of a larger body of work dedicated towards the judicious use of AI to optimize different elements within evolutionary process.

Additionally, I suggest that they include a comparison with deep learning-based methods (e.g., in response to comment 8) in their GitHub repository, which would provide readers with a clearer understanding of the approach.

We agree with the Reviewer and have provided additional information regarding the comparison of Kartezio with DL-based methods in the ReadMe file provided on the GitHub repository.